# Single-cell biological network inference using a heterogeneous graph transformer

Anjun Ma [1,2,7], Xiaoying Wang[3,7], Jingxian Li[3], Cankun Wang [1], Tong Xiao[2], Yuntao Liu[3], Hao Cheng[1], Juexin Wang [4,5], Yang Li[1], Yuzhou Chang[1,2], Jinpu Li[5,6], Duolin Wang [4,5], Yuexu Jiang[4,5], Li Su[5,6], Gang Xin[2], Shaopeng Gu[1], Zihai Li [2], Bingqiang Liu [3,8] ✉, Dong Xu [4,5,6,8] ✉ & Qin Ma [1,2,8] ✉

Single-cell multi-omics (scMulti-omics) allows the quantification of multiple modalities simultaneously to capture the intricacy of complex molecular mechanisms and cellular heterogeneity. Existing tools cannot effectively infer the active biological networks in diverse cell types and the response of these networks to external stimuli. Here we present DeepMAPS for biological network inference from scMulti-omics. It models scMulti-omics in a heterogeneous graph and learns relations among cells and genes within both local and global contexts in a robust manner using a multi-head graph transformer. Benchmarking results indicate DeepMAPS performs better than existing tools in cell clustering and biological network construction. It also showcases competitive capability in deriving cell-type-specific biological networks in lung tumor leukocyte CITE-seq data and matched diffuse small lymphocytic lymphoma scRNA-seq and scATAC-seq data. In addition, we deploy a DeepMAPS webserver equipped with multiple functionalities and visualizations to improve the usability and reproducibility of scMulti-omics data analysis.

Single-cell sequencing, such as single-cell RNA sequencing (scRNA-seq) and single-cell ATAC sequencing (scATAC-seq), reshapes the investigation of cellular heterogeneity and yields insights in neuroscience, cancer biology, immuno-oncology, and therapeutic responsiveness[1,2]. However, an individual single-cell modality only reflects a snapshot of genetic features and partially depicts the peculiarity of cells, leading to characterization biases in complex biological systems[2,3]. Single-cell multi-omics (scMulti-omics) allows the quantification of multiple modalities simultaneously to fully capture the intricacy of complex molecular mechanisms and cellular heterogeneity. Such analyses advance various biological studies when paired with robust computational analysis methods[4].

The existing tools for integrative analyses of scMulti-omics data, e.g., Seurat[5], MOFA+[6], Harmony[7], and totalVI[8], reliably predict cell types and states, remove batch effects, and reveal relationships or alignments among multiple modalities. However, most existing methods do not explicitly consider the topological information sharing among cells and modalities. Hence, they cannot effectively infer the active biological networks of diverse cell types simultaneously with cell clustering and have limited power to elucidate the response of these complex networks to external stimuli in specific cell types.

Recently, graph neural networks (GNN) have shown strength in learning low-dimensional representations of individual cells by propagating neighbor cell features and constructing cell-cell relations in a global cell graph[9,10]. For example, our in-house tool scGNN, a GNN

[1]Department of Biomedical Informatics, College of Medicine, The Ohio State University, Columbus, OH, USA. [2]Pelotonia Institute for Immuno-Oncology, The James Comprehensive Cancer Center, The Ohio State University, Columbus, OH, USA. [3]School of Mathematics, Shandong University, Jinan, Shandong, China. [4]Department of Electrical Engineering and Computer Science, University of Missouri, Columbia, MO, USA. [5]Christopher S. Bond Life Sciences Center, University of Missouri, Columbia, MO, USA. [6]Institute for Data Science and Informatics, University of Missouri, Columbia, MO, USA. [7]These authors contributed equally: Anjun Ma, Xiaoying Wang. [8]These authors jointly supervised this work: Bingqiang Liu, Dong Xu, Qin Ma. ✉e-mail: bingqiang@sdu.edu.cn; xudong@missouri.edu; qin.ma@osumc.edu

model, has demonstrated superior cell clustering and gene imputation performance based on large-scale scRNA-seq data[11]. Furthermore, a heterogeneous graph with different types of nodes and edges has been widely used to model a multi-relational knowledge graph[12]. It provides a natural representation framework for integrating scMulti-omics data and learning the underlying cell-type-specific biological networks. Moreover, a recent development in the attention mechanism for modeling and integrating heterogeneous relationships has made deep learning models explainable and enabled the inference of cell-type-specific biological networks[12,13].

In this work, we developed DeepMAPS (Deep learning-based Multi-omics Analysis Platform for Single-cell data), a heterogeneous graph transformer framework for cell-type-specific biological network inference from scMulti-omics data. This framework uses an advanced GNN model, i.e., heterogeneous graph transformer (HGT), which has the following advantages: (i) It formulates an all-in-one heterogeneous graph that includes cells and genes as nodes, and the relations among them as edges. (ii) The model captures both neighbor and global topological features among cells and genes to construct cell-cell relations and gene-gene relations simultaneously[9,14–16]. (iii) The attention mechanism in this HGT model enables the estimation of the importance of genes to specific cells, which can be used to discriminate gene contributions and enhances biological interpretability. (iv) This model is hypothesis-free and does not rely on the constraints of gene co-expressions, thus potentially inferring gene regulatory relations that other tools usually cannot find. It is noteworthy that DeepMAPS is implemented into a code-free, interactive, and non-programmatic interface, along with a Docker, to lessen the programming burden for scMulti-omics data.

## Results

### Overview of DeepMAPS

Overall, DeepMAPS is an end-to-end and hypotheses-free framework to infer cell-type-specific biological networks from scMulti-omics data. There are five major steps in the DeepMAPS framework (Fig. 1 and Methods). (i) Data are preprocessed by removing low-quality cells and lowly-expressed genes, and then different normalization methods are applied according to the specific data types. An integrated cell-gene matrix is generated to represent the combined activity of each gene in each cell. Different data integration methods are applied for different scMulti-omics data types[5–8]. (ii) A heterogeneous graph is built from the integrated matrix, including cells and genes as nodes and the existence of genes in cells as edges. (iii) An HGT model is built to jointly learn the low-dimensional embedding for cells and genes and generate an attention score to indicate the importance of a gene to a cell. (iv) Cell clustering and functional gene modules are predicted based on HGT-learned embeddings and attention scores. (v) Diverse biological networks, e.g., gene regulatory networks (GRN) and gene association networks, are inferred in each cell type.

To learn joint representations of cells and genes, we first generate a cell-gene matrix integrating the information of the input scMulti-omics data. A heterogeneous graph with cell nodes and gene nodes is then constructed, wherein an unweighted cell-gene edge represents the existence of gene activity of a gene in a cell, and the initial embedding of each node is learned from the gene-cell integrated matrix via two-layer GNN graph autoencoders (Methods). Such a heterogeneous graph offers an opportunity to clearly represent and organically integrate scMulti-omics data so that biologically meaningful features can be learned synergistically. The entire heterogeneous graph is then sent to a graph autoencoder to learn the relations between the cells and genes and update the embedding of each node. Here, DeepMAPS adopts a heterogeneous multi-head attention mechanism to model the overall topological information (global relationships) and neighbor message passing (local relationships) on the heterogeneous graph. The heterogeneous graph

representation learning provides a way to enable the embedding of cells and genes simultaneously using the transformer in DeepMAPS. The initial graph determines the path of message passing and how the attention scores can be calculated in DeepMAPS.

In each HGT layer, each node (either a cell or a gene) is considered a target, and its 1-hop neighbors as sources. DeepMAPS evaluates the importance of its neighbor nodes and the amount of information that can be passed to the target based on the synergy of node embedding (i.e., attention scores). As a result, cells and genes with highly positive correlated embeddings are more likely to pass messages within each other, thus maximizing the similarity and disagreement of the embeddings. To make the unsupervised training process feasible on a large heterogeneous graph, DeepMAPS is performed on 50 subgraphs sampled from the heterogeneous graph, covering a minimum of 30% of cells and genes to train for the shared parameters between different nodes, information which is later used for testing of the whole graph. As an important training outcome, an attention score is given to represent the importance of a gene to a cell. A high attention score for a gene to a cell implies that the gene is relatively important for defining cell identity and characterizing cell heterogeneity. This discrimination allows for the construction of reliable gene association networks in each cell cluster as the final output of DeepMAPS. We then build a Steiner Forest Problem (SFP) model[17] to identify genes with higher attention scores and similar embedding features to a cell cluster. The gene-gene and gene-cell relations in the optimized solution of the SFP model mirror the embedding similarity of genes and the attention importance of genes to each cell cluster. A gene association network can be established by genes with the highest importance in characterizing the identity of that cell cluster based on their attention scores and embedding similarities, and these genes are considered to be cell-type-active.

### DeepMAPS achieves superior performances in cell clustering and biological network inference from scMulti-omics data

We benchmarked the cell clustering performance of DeepMAPS on ten scMulti-omics datasets, including three multiple scRNA-seq datasets (R-bench-1, 2, and 3), three CITE-seq datasets (C-bench-1, 2, and 3), and four matched scRNA-seq and scATAC-seq (scRNA-ATAC-seq) datasets measured from the same cell (A-bench-1, 2, 3, and 4) (Supplementary Data 1). Specifically, the six R-bench and C-bench datasets have benchmark annotations provided in their original manuscripts, while the four A-bench datasets do not. These datasets cover the number of cells ranging from 3,009 to 32,029; an average read depth (considering scRNA-seq data only) ranging from 2,885 to 11,127; and a zero-expression rate (considering scRNA-seq data only) from 82 to 96% (Supplementary Data 1).

We compared DeepMAPS with four benchmarking tools (Seurat v3 and v4[5,18], MOFA + [6], TotalVI[8], Harmony[7], and GLUE[19] (Methods)) in terms of the Average Silhouette Width (ASW), Calinski-Harabasz (CH), Davies-Bouldin Index (DBI), and Adjusted Rand Index (ARI) to evaluate cell clustering performance. For each dataset, we trained DeepMAPS on 36 parameter combinations, including the number of heads, learning rate, and the number of training epochs. To ensure fairness, each benchmarking tool was also tuned with different parameter combinations (Methods). DeepMAPS achieves the best performance comparing all benchmark tools in all test datasets in terms of ARI (for R-benches and C-benches) and ASW (for A-benches) (Fig. 2a, Supplementary Figs. 1–3, and Source Data 1-3). We also noticed that Seurat was the second-best performed tool, with small variances for different parameter selections in all benchmark datasets. We selected the default parameter per data type based on the performance of parameter combinations on the grid-search benchmarking. The parameter combination with the highest median ARI/ASW scores averaged in all benchmark datasets was considered as the default parameters for the corresponding data type.

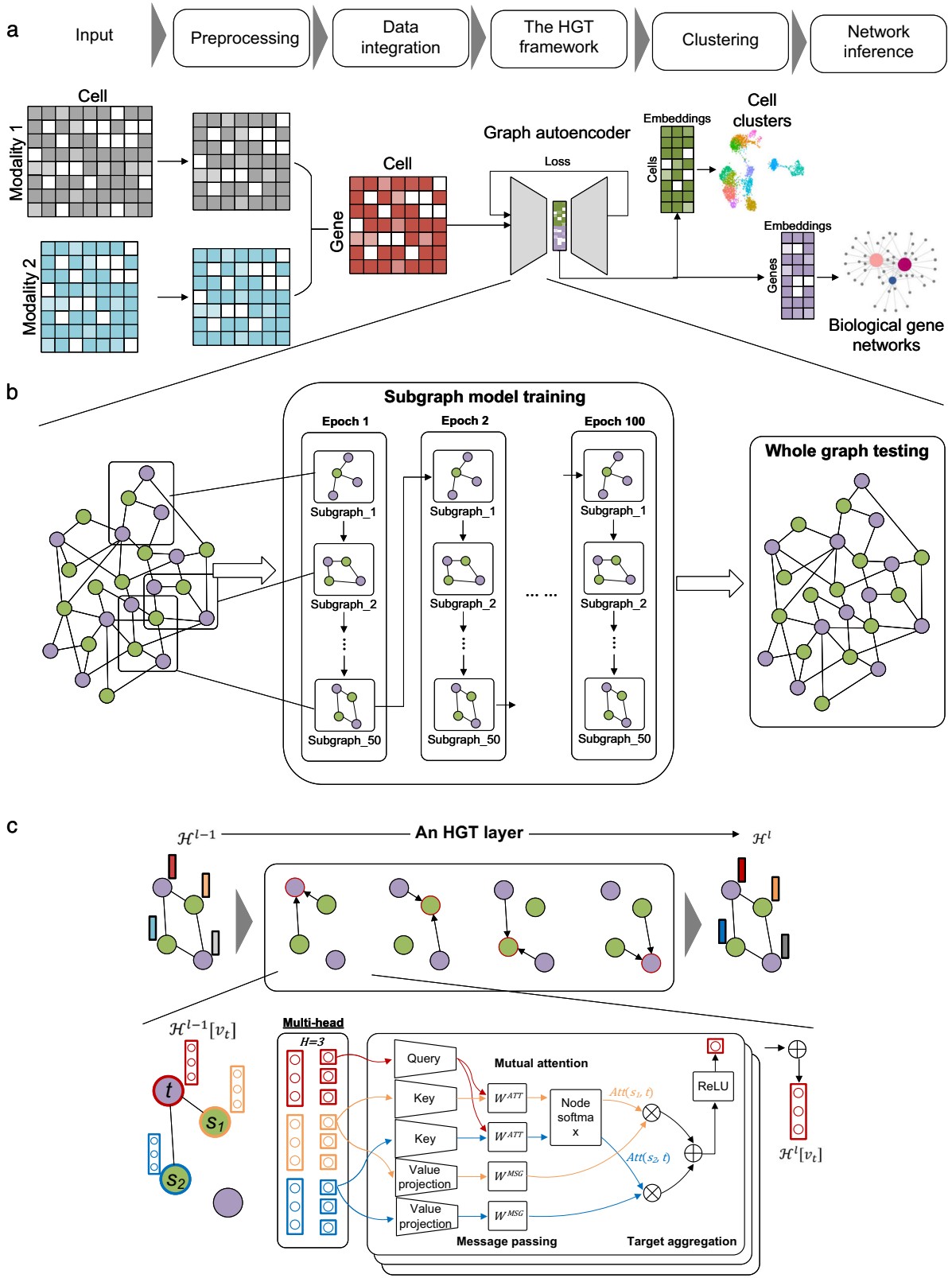

Additional benchmarking experiments were carried out to justify the selection of different integration methods in DeepMAPS. Specifically, for the analysis of scRNA-ATAC-seq data, we designed an integration method using gene velocity to balance the weight between gene expressions and chromatin accessibilities in characterizing cell activities and states (Methods). This integration process can ensure harmonizing datasets (especially for multiple scRNA-seq data) and generate an integrated matrix (with genes as rows and cells as columns) as the input for HGT. Our results showed that, for benchmark data 1 and 2 (A-bench-1 and −2), the velocity-based approach showed significantly ($p$-value <0.05) higher ASW scores than the weighted nearest neighbor (WNN) approach in Seurat v 4.0 on all grid-search parameter combinations (Supplementary Fig. 4 and Source Data 4). We reason that with the inclusion of velocity information, the modality weight between gene

**Fig. 1 | The workflow of DeepMAPS and HGT illustration. a** The overall framework of DeepMAPS. Five main steps were included in carrying out cell clustering and biological gene network inference from the input scMulti-omics data. **b** The graph autoencoder was inserted with a HGT model. The integrated cell-gene matrix was used to build a heterogeneous graph include all cells (green) and genes (purple) as nodes. The HGT model is trained on multiple subgraphs (50 subgraphs as an example) that cover nodes in the whole graph as many as possible. Each subgraph is used to train the model with 100 epochs; thus, the whole training process iterates 5,000 times. The trained model is then applied to the whole graph to learn and update the embeddings of each node. **c** An illustration of embedding update process of the target node in a single HGT layer. The red circle in the upper panel indicates the target node and the black circle indicates the source nodes. Arrows represents for the connection between a target node and source nodes. Colored rectangles represent for embeddings of different nodes. The zoom in detailed process in the bottom panel shows the massage passing process and attention mechanism. The final output of one HGT layer is an update of node embedding for all nodes. HGT heterogeneous graph transformer.

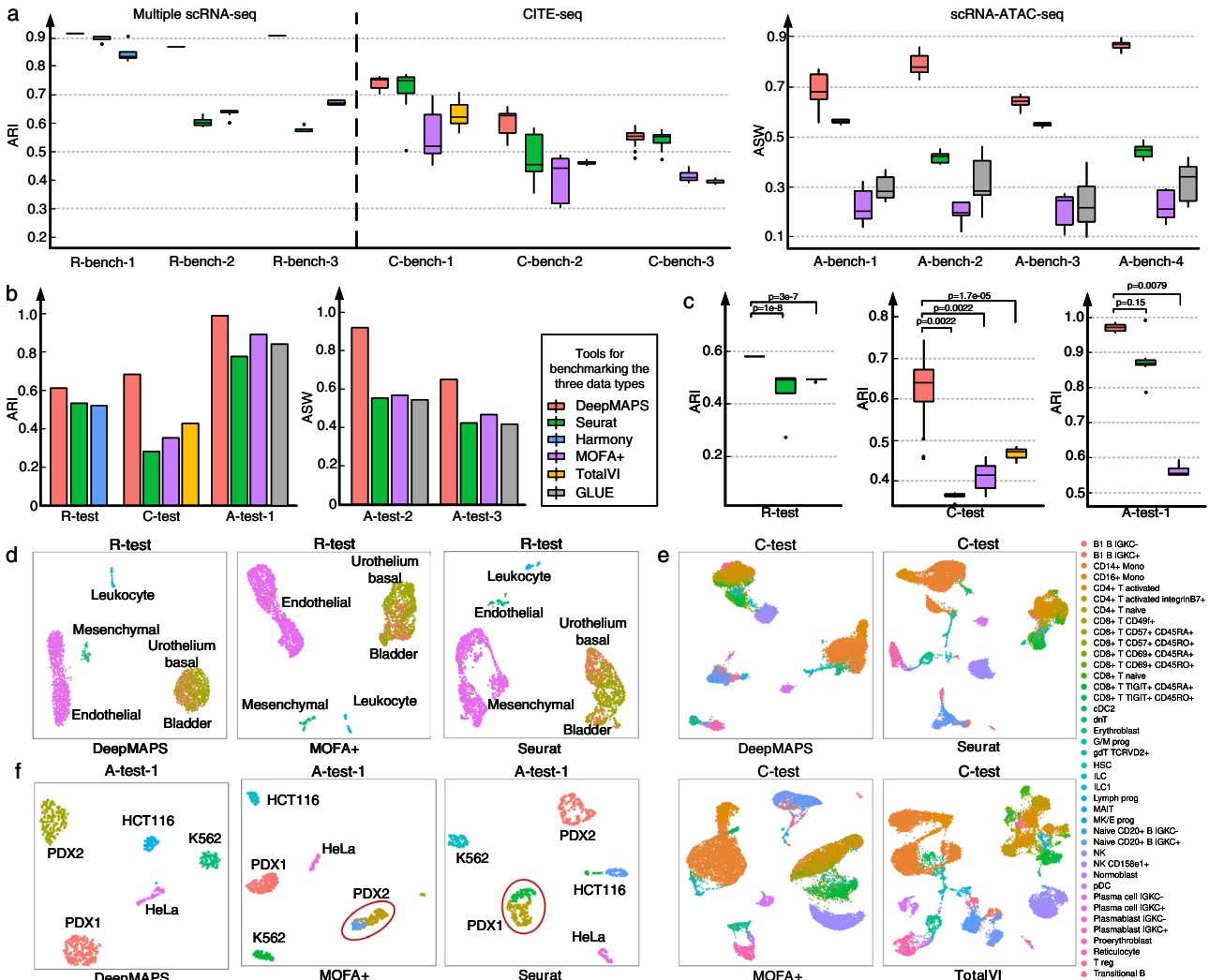

**Fig. 2 | Benchmarking of DeepMAPS in terms of cell clustering. a** Benchmark cell clustering results of ten datasets in ARI for the three multiple scRNA-seq data and the three CITE-seq data with benchmark labels, and ASW for the four scRNA-ATAC-seq data without benchmark labels. Each box showcases the minimum, first quartile, median, third quartile, and maximum ARI or AWS results of a tool using different parameter settings (DeepMAPS: $n = 96$, Seurat: $n = 16$ for RNA-RNA and CITE-seq and 36 for RNA-ATAC, Harmony: $n = 36$, MOFA + : $n = 36$, TotalVI: $n = 48$, and GLUE: $n = 72$). Dots represent outliers. **b** Results comparison on five independent datasets. No repeated experiment was conducted. **c** Robustness test of DeepMAPS using the cell cluster leave-out method for the three independent test datasets with benchmarking cell labels. *p*-values were calculated based on two-tail t.test. Each box showcases the minimum, first quartile, median, third quartile, and maximum ARI results of a tool performed on different data subsets (R-test: $n = 5$, C-test: $n = 20$, and A-test-1: $n = 5$). Dots represent outliers. **d**–**f** UMAP comparison of R-test, C-test, and A-test-1 datasets between DeepMAPS and other tools using the original cell labels. Source data are provided as a Source Data file. ASW average Silhouette width, ARI adjusted rand index.

expression and chromatin accessibility that contribute to recognize cell types are better balanced (Supplementary Fig. 5). The comparison of modality weight of scATAC-seq in different cell clusters by using or without using the velocity-weighted balance method. In addition, we compared different clustering methods (i.e., Leiden, Louvain, and SLM) in DeepMAPS and compared the impact of clustering resolutions (i.e., 0.4, 0.8, 1.2, and 1.6) to cell clustering results. We found no significant

differences among these clustering methods, and Louvain showed slightly better performance than the other two (Supplementary Fig. 6 and Source Data 5). Lastly, DeepMAPS achieved higher scores than other tools when selecting the same clustering resolution. We also found that, in most cases, higher resolution lower cell clustering prediction scores; therefore, we selected resolution at 0.4 as the default parameter in DeepMAPS (Supplementary Fig. 7 and Source Data 6–8).

We further independently tested our default parameter selection on five independent datasets, named R-test, C-test, A-test-1, −2, and −3, by comparing our results with the same benchmarking tools using their default parameters. For the three test datasets with benchmarking cell labels, DeepMAPS performs the best in terms of ARI score, while for the two scRNA-ATAC-seq datasets without cell labels, the benchmarking tools in the comparison achieve similar performance (Fig. 2b and Source Data 9). In order to evaluate the robustness of DeepMAPS, a leave-one-out test was performed on the three independent test datasets with benchmark labels (Fig. 2c and Source Data 10). We first removed all cells in a cell cluster based on benchmark labels and then applied DeepMAPS and other tools on the remaining cells. For each dataset, the leave-one-out results of DeepMAPS were better than the other tools with higher ARI scores, indicating that the message passing and attention mechanism used in DeepMAPS maintains cell-cell relations in a robust manner.

The cell clustering UMAP on the three independent datasets with benchmarking labels showcased that the latent representations obtained in DeepMAPS can better preserve the heterogeneity of scRNA-seq data (Fig. 2d–f). For the R-test dataset, all tools showed the

ability to separate mesenchymal, leukocyte, and endothelial cells, but failed to separate urothelium basal cells and bladder cells. However, cells on the DeepMAPS UMAP are more compact, and the bladder cells (red dots) are grouped better than MOFA + and Seurat (Fig. 2d). For the C-test dataset, cells in the same cluster are more ordered and compact (e.g., the B cell cluster and NK cell cluster), while cells from different clusters are more apart from each other on DeepMAPS UMAP (e.g., CD8 cell clusters and CD4 cell clusters). (Fig. 2e). For the A-test-1 dataset, DeepMAPS was the only tool that accurately separated each cell type. In contrast, Seurat and MOFA + mistakenly divided the PDX1 or PDX2 population into two clusters and included more mismatches (Fig. 2f).

**DeepMAPS can infer statistically significant and biologically meaningful gene association networks from scMulti-omics data**
We evaluated the two kinds of biological networks that DeepMAPS can infer, gene association network and GRN, in terms of centrality scores and functional enrichment. For the R-test dataset (Fig. 3a) and C-test dataset (Fig. 3b), we used two centrality scores, closeness centrality (CC) and eigenvector centrality (EC), that have been used in previous

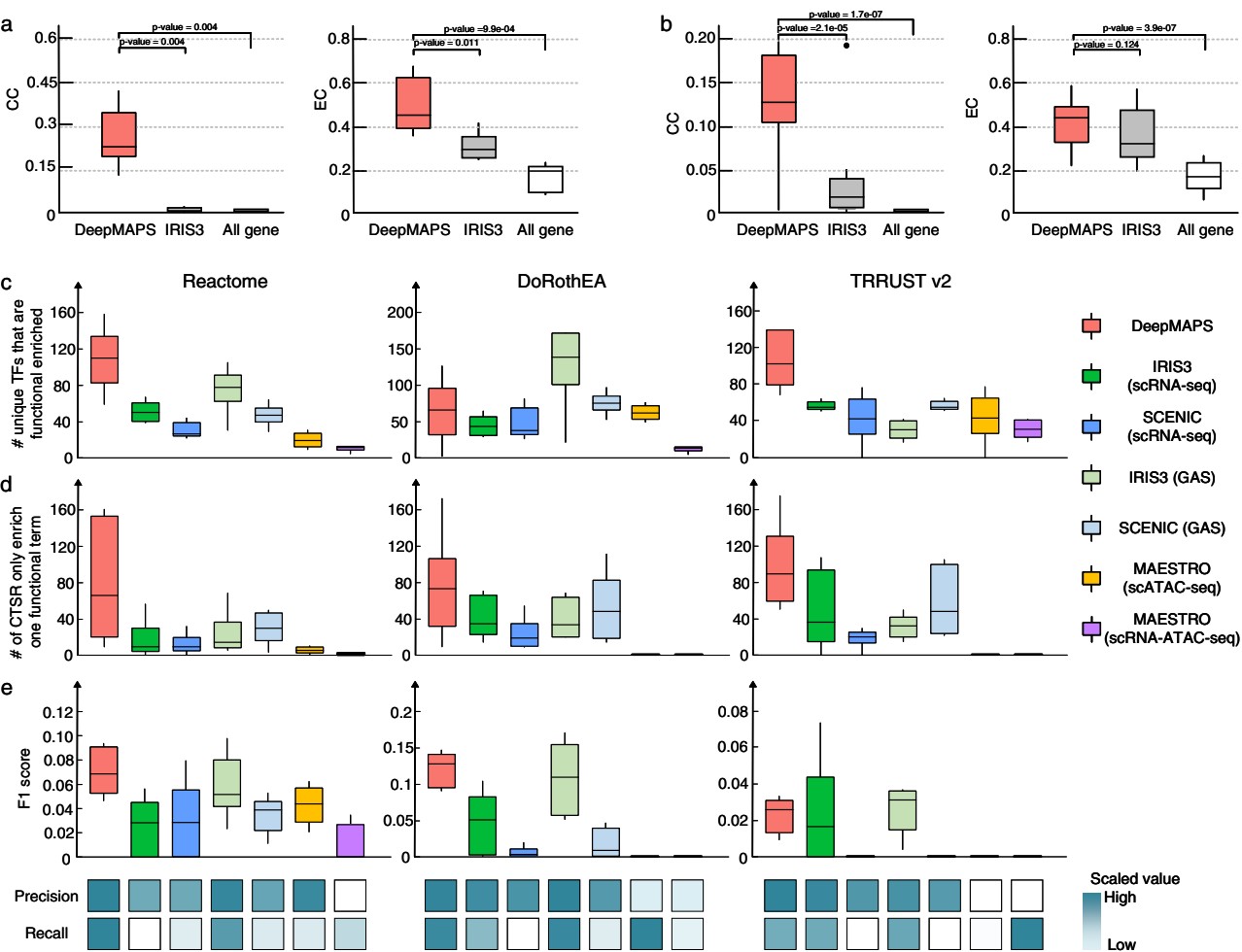

**Fig. 3 | Evaluation and comparison of gene association network inference of DeepMAPS. a, b** Closeness centrality (CC) and eigenvector centrality (EC) were used to indicate the compactness and importance of genes to the network. We compared our results with IRIS3 and a background network using all genes for the R-test dataset ($n = 5$) **a** and C-test dataset ($n = 14$) **b**. $p$-values were calculated using a two-tail t-test. **c** Comparison of the number of unique TFs in GRNs that showed significantly enriched biological functions in three public databases. Each box contains the results of six scRNA-ATAC-seq datasets ($n = 6$). **d** Comparison of the number of cell-type-specific regulons in GRNs significantly enriched in only one

biological function/pathway in the three public databases ($n = 6$). **e** The F1 score comparisons of regulons enriched to only one function/pathway using three databases ($n = 6$). The mean value of precision and recall scores of the selected six scRNA-ATAC-seq datasets were max-min scaled and shown in the heatmap with darker blue indicating high values and lighter blue indicating low values. Source data are provided as a Source Data file. Each box in Fig. 3 showcases the minimum, first quartile, median, third quartile, and maximum score of the corresponding criteria. CC closeness centrality, EC eigenvector centrality, CTSR cell-type-specific regulon.

single-cell gene association network evaluations[20], to compare the identified gene association networks from all the tools in this comparison. CC reflects the average connectivity of a node to all others in a network, and EC reflects the importance of a node based on its connected nodes. Both CC and EC can interpret node's influence in identifying genes that may play more critical roles in the network. A gene association network with higher node centrality indicates that the detected genes are more likely to be involved in critical and functional biological systems. We also constructed gene co-expression networks as a background using all genes in a dataset by calculating Pearson's correlation coefficients of gene expression in a cell cluster. $p$-value = 0.05 was set as the edge cutoff. We compared cell-type-active gene association networks generated in DeepMAPS with those generated in IRIS3[15] and the background co-expression networks. The average CC and EC of networks constructed by DeepMAPS in R-test and C-test datasets showed significantly higher scores than IRIS3 and the background co-expression networks (Source Data 11). We reason that the gene association network generated in DeepMAPS is not only co-expressed but also of great attention impact on cells; thus, genes in the network tend to be more important to the cell type.

To evaluate whether DeepMAPS can identify biologically meaningful GRNs in specific cell types, we performed enrichment tests on basic gene regulatory modules (i.e., regulons[14]), with three public functional databases, Reactome[21], DoRothEA[22], and TRRUST v2[23]. To avoid any bias in comparison, we compared cell-type-specific GRNs inferred from DeepMAPS with (i) IRIS3 and SCENIC[14] on the scRNA-seq matrix, (ii) IRIS3 and SCENIC on a gene-cell matrix recording the gene activity scores (GAS) calculated in DeepMAPS based on the velocity-based integration method, (iii) MAESTRO[24] on scATAC-seq matrix, and (iv) MAESTRO on original scRNA-seq and scATAC-seq matrix. The six datasets collected from human tissue were used (i.e., A-test-1, A-bench-2, A-bench-3, A-bench-4, A-test-1, A-test-2). We first showcased that the GRNs identified in DeepMAPS included more unique transcription factor (TF) regulations than the other tools, except for the enrichment to the DoRothEA database (Fig. 3c and Source Data 12). We considered that a highly cell-type-specific regulon (CTSR) might represent only one significant enriched functionality; alternatively, a generic regulon might improperly contain genes involved in several pathways. Therefore, we compared the number of CTSRs enriched to one function/pathway across different tools. DeepMAPS outperformed ($p$-value<0.05) other tools on most of the six scRNA-ATAC-seq datasets in terms of the number of regulons that enrich only one function/pathway and the enrichment F1 scores (Fig. 3d, e and Source Data 12). For the F1 score of the enrichment test to the TRRUST v2 database, DeepMAPS (median F1 score is 0.026) was slightly lower than IRIS3 using the GAS matrix (median F1 score is 0.031). We also noticed that all tools did not achieve good enrichments in the TRRUST v2 database mainly due to the small number of genes (on average, 10 genes are regulated by one TF; 795 TFs in total). SCENIC also showed competitive scaled precision scores (scaled mean: 0.47 for Reactome, 0.66 for DoRothEA, and 0.61 for TRRUST v2), while achieving lower scaled recall scores, making the F1 scores smaller than DeepMAPS for most datasets. IRIS3 and SCENIC performed on the GAS matrix showed better enrichment results than using scRNA-seq data only, indicating that integrating information from scRNA-ATAC-seq data is more useful for GRN inference than using scRNA-seq data alone.

## DeepMAPS accurately identifies cell types and infers cell-cell communication in PBMC and lung tumor immune CITE-seq data

We present a case study that applies DeepMAPS to a published mixed peripheral blood mononuclear cells (PBMC) and lung tumor leukocytes CITE-seq dataset (10× Genomics online resource, Supplementary Data 1) to demonstrate capacity in modeling scMulti-omics in characterizing cell identities. The dataset includes RNAs and proteins measured on 3485 cells. DeepMAPS identified 13 cell clusters,

including four CD4+ T cell groups (naive, central memory (CM), tissue-resident memory (TRM), and regulatory (Treg)), two CD8+ T cell groups (CM and TRM), a natural killer cell group, a memory B cell group, a plasma cell group, two monocyte groups, one tumor-associated macrophage (TAM) group, and a dendritic cell (DC) group. We annotated each cluster by visualizing the expression levels of curated maker genes and proteins (Fig. 4a, b and Supplementary Data 2). Compared to cell types identified using only proteins or RNA, we isolated or accurately annotated cell populations that could not be characterized using the individual modality analysis. For example, the DC cluster was only successfully identified using the integrated protein and RNA. By combining signals captured from both RNA and proteins, DeepMAPS successfully identified biologically reasonable and meaningful cell types in the CITE-seq data.

We then compared the modality correlation between the two cell types. We used the top differentially expressed genes and proteins between memory B cells and plasma cells, and performed hierarchical clustering of the correlation matrix. The result clearly stratified these features into two anticorrelated modules: one associated with memory B cells and the other with plasma cells (Fig. 4c). Furthermore, we found that the features in the two modules significantly correlated with the axis of maturation captured by our HGT embeddings (Supplementary Fig. 8 and Supplementary Data 3). For example, one HGT embedding (the 51st) showed distinctive differences between plasma cells and memory B cells (Fig. 4d, e). Similar findings were also observed when comparing EM CD8+ T cells with TRM CD8+ T cells (Fig. 4f). Nevertheless, it is possible to identify a representative HGT embedding (56th) that maintains embedding signals for a defined separation of the two groups (Fig. 4g, h). These results point to any two cell clusters consisting of coordinated activation and repression of multiple genes and proteins, leading to a gradual transition in cell state that can be captured by a specific dimension of the DeepMAPS latent HGT space. On the other hand, we generated the gene-associated networks with genes showing high attention scores for EM CD8+ T cells, TRM CD8+ T cells, memory B cells, and plasma cells and observed diverse patterns (Supplementary Fig. 9).

Based on the cell types and raw data of gene and protein expressions, we inferred cell–cell communications and constructed communication networks among different cell types within multiple signaling pathways using CellChat[25] (Fig. 4i). For example, we observed a CD6-ALCAM signaling pathway existing between DC (source) and TRM CD4+ T cells (target) in the lung cancer tumor microenvironment (TME). Previous studies have shown that ALCAM on antigen-presenting DCs interacts with CD6 on the T cell surface and contributes to T cell activation and proliferation[26–28]. As another example, we identified the involvement of the NECTIN-TIGIT signaling pathway during the interaction between the TAM (source) and TRM CD8+ T cells (target), which is supported by a previous report that NECTIN (*CD155*) expressed on TAM could be immunosuppressive when interacting with surface receptors, TIGIT, on CD8+ T cells in the lung cancer TME[29,30].

## DeepMAPS identifies specific GRNs in diffuse small lymphocytic lymphoma scRNA-seq and scATAC-seq data

To further extend the power of DeepMAPS to GRN inference, we used a single-cell Multiome ATAC + Gene expression dataset available on the 10× Genomics website (10× Genomics online resource). The raw data is derived from 14,566 cells of flash-frozen intra-abdominal lymph node tumor from a patient diagnosed with diffuse small lymphocytic lymphoma (DSLL) of the lymph node lymph. We integrate gene expression and chromatin accessibility by balancing the weight of each modality of a gene in a cell based on RNA velocity (Fig. 5a and Method). To build TF-gene linkages, we considered gene expression, gene accessibility, TF-motif binding affinity, peak-to-gene distance, and TF-coding gene expression. Genes found to be

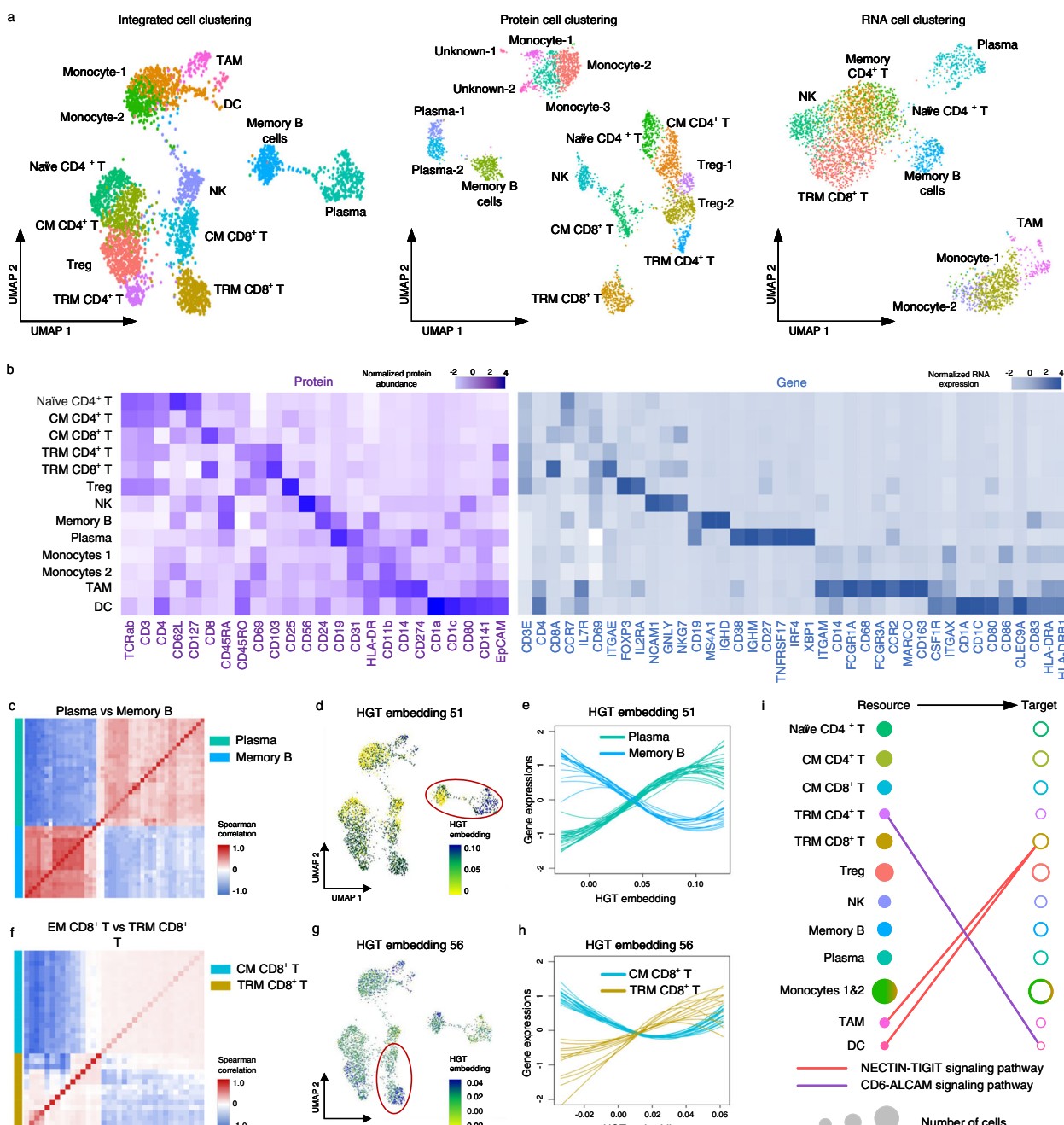

**Fig. 4 | DeepMAPS identification of heterogeneity in CITE-seq data of PBMC and lung tumor leukocytes. a** UMAPs for DeepMAPS cell clustering results from integrated RNA and protein data, protein data only, and RNA data only. Cell clusters were annotated based on curated marker proteins and genes. **b** Heatmap of curated marker proteins and genes that determine the cell clustering and annotation. **c** Heatmap of the Spearman correlation comparison of top differentially expressed genes and proteins in plasma cells and memory B cells. **d** UMAP is colored by the 51st embedding, indicating distinct embedding representations in plasma cells and memory B cells. **e** Expression of top differentially expressed genes and proteins in *c* as a function of the 51st embedding to observe the pattern relations between plasma cells and memory B cells. Each line represents a gene/protein, colored by cell types. For each gene, a line was drawn using a loess smoothing function based on the

corresponding embedding and scaled gene expression in a cell. **f–h** Similar visualization was conducted for the 56th embedding to compare EM CD8+ T cells and TRM CD8+ T cells *c–e*. **i** Two signaling pathways, NECTIN and ALCAM, are shown to indicate the predicted cell–cell communications between two cell clusters. A link between a filled circle (resource cluster with highly expressed ligand coding genes) and an unfilled circle (target cluster with highly expressed receptor coding genes) indicates the potential cell-cell communication of a signaling pathway. Circle colors represent different cell clusters, and the size represents the number of cells. The two monocyte groups were merged. TRM tissue-resident memory, CM central memory, TAM tumor-associated macrophage, HGT heterogeneous graph transformer.

regulated by the same TF in a cell cluster are grouped as a regulon. We considered regulons with higher centrality scores to have more significant influences on the characterization of the cell cluster. Regulons regulated by the same TF across different cell clusters are compared for differential regulon activities. Those with significantly

higher regulon activity scores (RAS) are considered as the cell-type-specific regulons in the cell cluster.

DeepMAPS identified 11 cell clusters in the DSLL data. All clusters were manually annotated based on curated gene markers (Fig. 5b and Supplementary Data 4). Two DSLL-like cell clusters (DSLL state-1 and

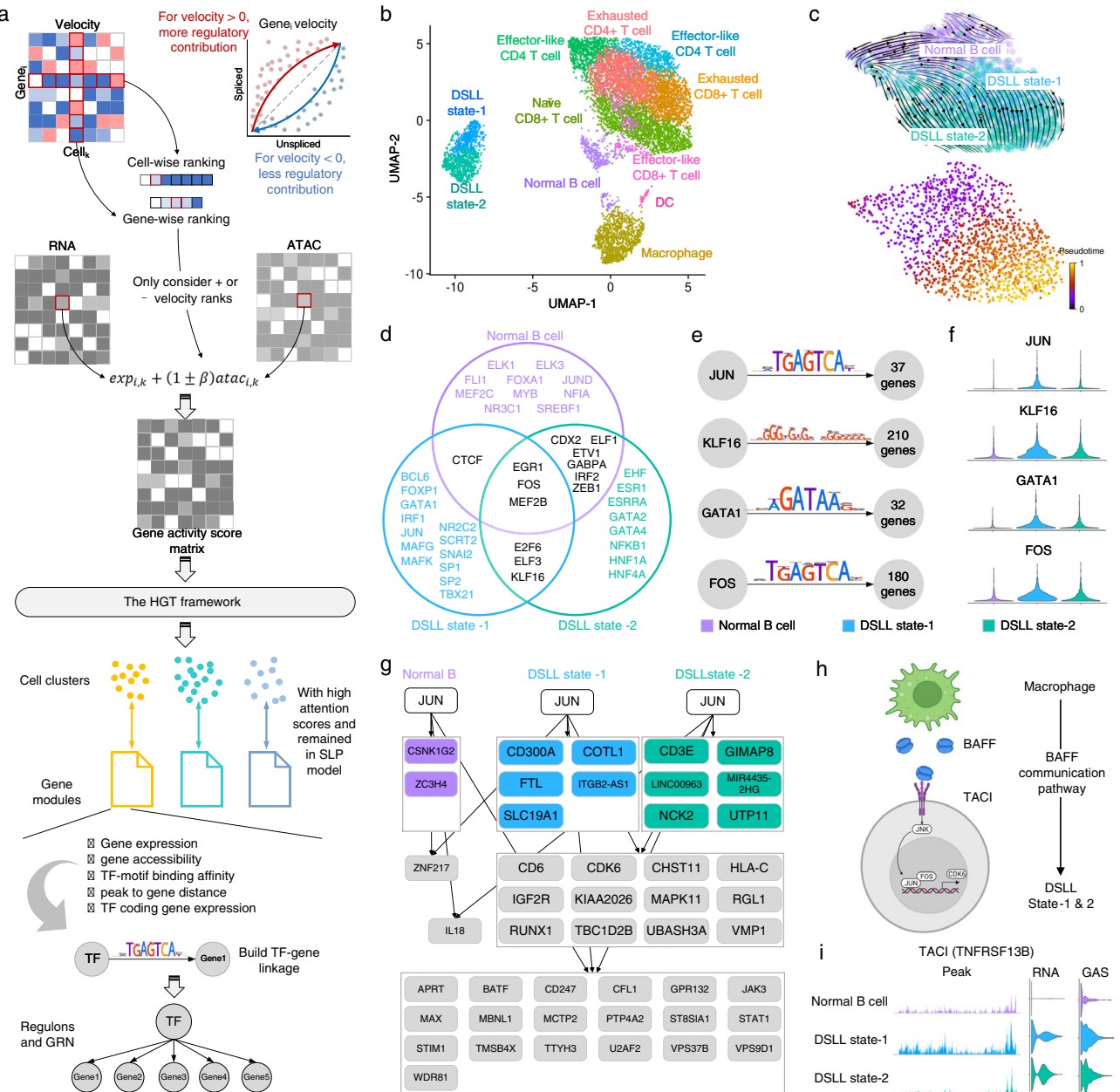

**Fig. 5 | DeepMAPS identifies specific GRNs in DSLL subnetworks. a** Conceptual illustration of DeepMAPS analysis of scRNA-ATAC-seq data. Modalities are first integrated based on a velocity-weighted balance. The integrated GAS matrix was then used to build a heterogeneous graph as input into the HGT framework. The cell cluster and gene modules with high attention scores were then used for building TF-gene linkages and determining regulons in each cell cluster. **b** The UMAP shows the clustering results of DeepMAPS. Cell clusters were manually annotated based on curated marker genes. **c** The observed and extrapolated future states (arrows) based on the RNA velocity of the normal B cell and the two DSLL states are shown (top panel). Velocity-based trajectory analysis shows the pseudotime from the top to the bottom right (bottom panel). **d** Selected 20 TF in each of the three clusters, representing the top 20 regulons with the highest centrality scores. Colors represent regulons uniquely identified in each cluster or shared between different clusters. **e** Regulons in DSLL state-1 showed a significant difference in regulon activity compared to the other clusters. Motif shape and number of regulated genes are also shown. **f** Violin plots of regulon activities of the four regulons compared between the three clusters. **g** The downstream-regulated genes of JUN (the most differentially active regulon in DSLL state-1) in the three clusters. **h** An illustration of the BAFF signaling pathway identified from GAS-based cell-cell communication prediction using CellChat. The BAFF signaling pathway was found to exist between macrophage and both DSLL states. It further activates the JUN regulon and enables the transcription of genes like *CDK6*. Figure created with BioRender.com. **i** The ATAC peak, RNA expression, and GAS level of *TNFRSF13B* (the coding gene of TACI, the receptor in the BAFF signaling pathway). Source data are provided as a Source Data file.

state-2) were observed. The RNA velocity-based pseudotime analysis performed on the three B cell clusters (normal B cell and two DSLL states) assumed that the two DSLL states were derived from normal B cells, and state-1 was derived earlier than state-2, although the two states seemed to be partially mixed (Fig. 5c). We further selected the top 20 TFs with the highest regulon centrality scores in each of the

three cell clusters (Fig. 5d and Source Data 13). Interestingly, these TFs showed distinctions between the normal and the two DSLL states and inferred variant regulatory patterns within the two DSLL states. For regulons shared by all three B cell clusters, EGR1, MEF2B, and FOS were transcriptionally active in both normal B and DSLL cells and responsible for regulating B cell development, proliferation, and germinal

center formation[31–34]. E2F6, ELF3, and KLF16 were identified as shared only in the two DSLL states, with reported roles in tumorigenesis[35–40]. Further, JUN, MAFK, and MAFG, which encode the compartments of the activating protein-1 (AP-1),[34,41,42] were found to be active in DSLL state-1 while NFKB1, coding for a subunit of the NF-κB protein complex[43,44], was found to be active in DSLL state-2.

We constructed a GRN consisting of the four cell-type-specific regulons (JUN, KLF16, GATA1, and FOS) (Fig. 5e and Supplementary Fig. 10) in DSLL state-1 with RAS that is significantly higher than normal B cells and DSLL state-2 (Fig. 5f). KLF16 reportedly promotes the proliferation of both prostate[39] and gastric cancer cells[40]. FOS and JUN are transcription factors in the AP-1 family, regulating the oncogenesis of multiple types of lymphomas[34,41,42,45], and GATA1 is essential for hematopoiesis, the dysregulation implicated in multiple hematologic disorders, and malignancies[46,47]. Distinct regulatory patterns were also observed when we zoomed in on a single regulon (Fig. 5g and Supplementary Figs. 11-12). As the most active regulon in DSLL state-1, JUN was found to regulate five unique downstream genes and 12 genes shared with DSLL state-2. Downstream genes, including *CDK6*[33,34], *IGF2R*[48], and *RUNX1*[49], are critical for cell proliferation, survival, and development functions in DSLL.

Moreover, we further built connections between upstream cell-cell communication signaling pathways and downstream regulatory mechanisms in DSLL cells. We identified a cell-cell communication between macrophage and the two DSLL states via the B cell activation factor (BAFF) signaling pathway, based on the integrated GAS matrix using CellChat[25], which includes BAFF as the ligand on macrophage cells and TACI (transmembrane activator and calcium-modulator and cyclophilin ligand interactor) as the receptor on DSLL cells (Fig. 5h). BAFF signaling is critical to the survival and maturation of normal B cells[50,51], while aberrations contribute to the resistance of malignant B cells to apoptosis[52,53]. We observed that the expression of the TACI coding gene, *TNFRSF13B*, was explicitly higher in the two DSLL states, while the corresponding chromatin accessibility maintained high peaks in state-1 (Fig. 5i). Upon engagement with its ligand, TACI has been reported to transduce the signal and eventually activate the AP-1[54,55] and NF-κB[56,57] transcriptional complexes for downstream signaling in B cells. JUN (a subunit of AP-1) was identified as the most specific and key regulator in state-1 responsible for cell proliferation and regulating downstream oncogenes, such as *CDK6*, that has been reported to promote the proliferation of cancer cells in multiple types of DSLLs as well as other hematological malignancies[58–60]. It is clear that BAFF signaling first appears in DSLL state-1 and triggers the activation of the JUN regulatory mechanism, leading to a high regulon activity of JUN. The JUN regulon accelerates the proliferation and oncogenesis explicitly in DSLL, leading to a more terminal differential stage of DSLL (state-2). As a result, state-1 includes cells undergoing rapid cell proliferation and differentiation, transitioning from normal B cells to matured DSLL. In short, DeepMAPS can construct GRNs and identify cell-type-specific regulatory patterns to offer a better understanding of cell states and developmental orders in diseased subpopulations.

### DeepMAPS provides a multi-functional and user-friendly web portal for analyzing scMulti-omics data

Due to the complexity of single-cell sequencing data, more webservers and dockers have been developed in the past three years[61–73] (Supplementary Data 5). However, most of these tools only provide minimal functions such as cell clustering and differential gene analysis. They do not support the joint analysis of scMulti-omics data and especially lack sufficient support for biological network inference. On the other hand, we recorded the running time of DeepMAPS and benchmark tools on different datasets with cell numbers ranging from 1000 to 160,000 (Supplementary Data 6). The deep learning models (DeepMAPS and TotalVI) have longer running time than Seurat and MOFA + . To these ends, we provided a code-free, interactive, and non-programmatic interface to lessen the programming burden for scMulti-omics data (Fig. 6a). The webserver supports the analysis of multiple RNA-seq data, CITE-seq data, and scRNA-ATAC-seq data using DeepMAPS (Fig. 6b). Some other methods, e.g., Seurat, are also incorporated as an alternative approach for the users' convenience. Three major steps—data preprocessing, cell clustering and annotation, and network construction—are included in the server. In addition, the DeepMAPS server supports real-time computing and interactive graph representations. Users may register for an account to have their own workspace to store and share analytical results. Other than the advances mentioned, the DeepMAPS webserver highlights an additional function for the elucidation of complex networks in response to external stimuli in specific cell types. The user can upload a metadata file with phenotype information (e.g., cells with treatment and without treatment), select, and re-label the corresponding cells (e.g., CD8+ T cells with treatment and CD8+ T cells without treatment). In this way, DeepMAPS will predict the treatment-related networks in CD8+ T cells. Examples are given in the online tutorial at https://bmblx.bmi.osumc.edu/tutorial.

## Discussion

DeepMAPS is a deep-learning framework that implements heterogeneous graph representation learning and a graph transformer in studying biological networks from scMulti-omics data. By building a heterogeneous graph containing both cells and genes, DeepMAPS identifies their joint embedding simultaneously and enables the inference of cell-type-specific biological networks along with cell types in an intact framework. Furthermore, the application of a heterogeneous graph transformer models the cell-gene relation in an interpretable uniform multi-relation. In such a way, the training and learning process in a graph can be largely shortened to consider cell impacts from a further distance.

By jointly analyzing gene expression and protein abundance, DeepMAPS accurately identified and annotated 13 cell types in a mixed CITE-seq data of PBMC and lung tumor leukocytes based on curated markers that cannot be fully elucidated using a single modality. We have also proved that the embedding features identified in DeepMAPS capture statistically significant signals and amplify them when the original signals are noisy. Additionally, we identified biologically meaningful cell-cell communication pathways between DC and TRM CD4+ T cells based on the gene association network inferred in the two clusters. For scRNA-ATAC-seq, we employed an RNA velocity-based method to dynamically integrate gene expressions and chromatin accessibility that enhanced the prediction of cell clusters. Using this method, we identified distinct gene regulatory patterns among normal B cells and two DSLL development states. We further elucidated the deep biological connections between cell-cell communications and the downstream GRNs, which helped characterize and define DSLL states. The identified TFs and genes can be potential markers for further validation and immuno-therapeutical targets in DSLL treatment.

While there are advantages and improved performances for analyzing scMulti-omics data, there is still room to improve the power of DeepMAPS further. First, the computational efficiency for super-large datasets (e.g., more than 1 million cells) might be a practical issue considering the complexity of the heterogeneous graph representation (which may contain billions of edges). Moreover, DeepMAPS is recommended to be run on GPUs, which leads to a potential problem of reproducibility. Different GPU models have different floating-point numbers that may influence the precision of loss functions during the training process. For different GPU models, DeepMAPS may generate slightly different cell clustering and network results. Lastly, the current version of DeepMAPS is based on a bipartite heterogeneous graph with genes and cells. Separate preprocessing and integration steps are required to transfer different modalities to genes for integration into a cell-gene matrix. To fully achieve an end-to-end framework for

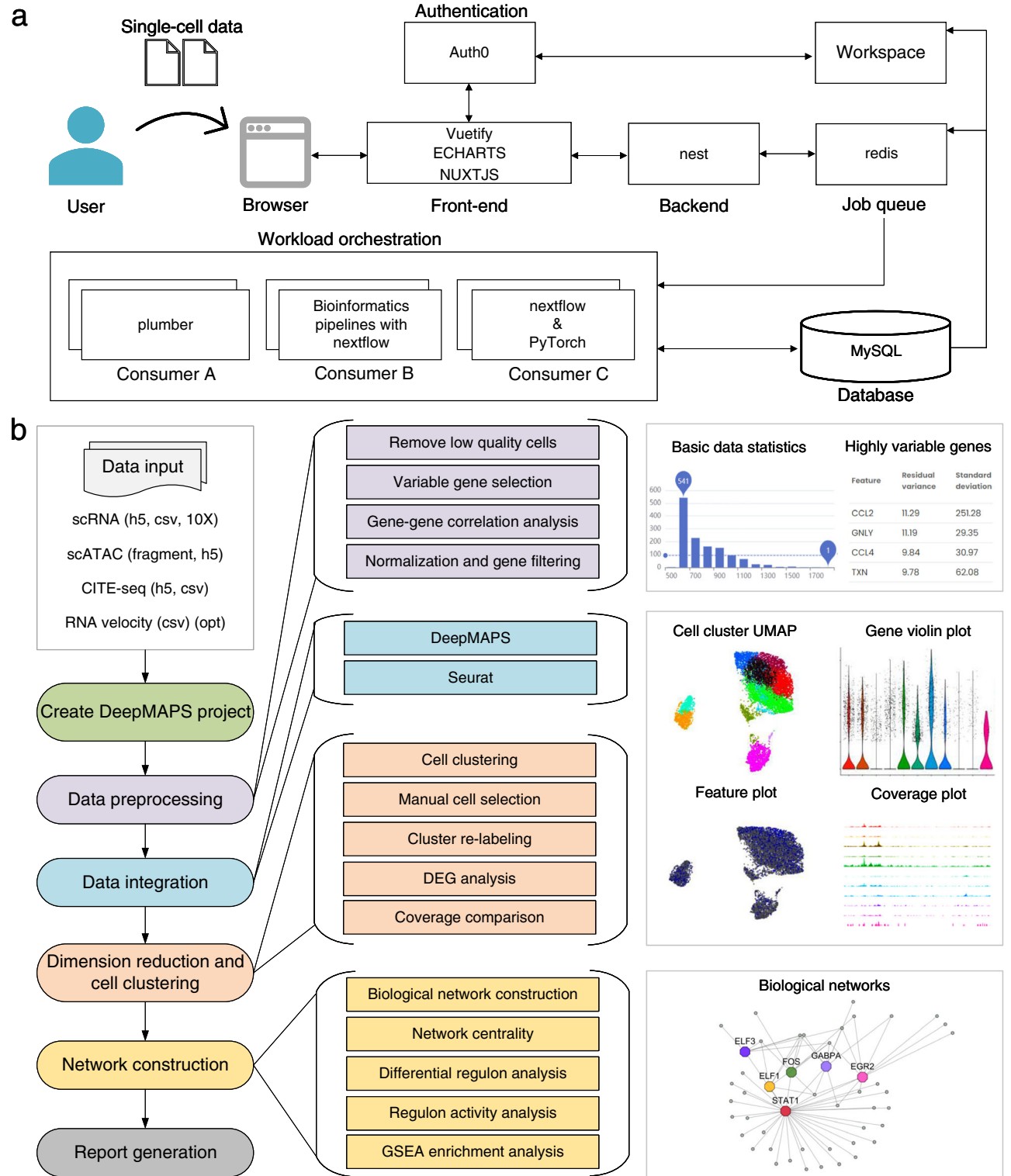

**Fig. 6 | The organization of the DeepMAPS web portal. a** Software-engineering diagram of DeepMAPS and an overview of the framework. **b** Pipeline illustration of the server, including major steps (left; colors indicate different steps), detailed analyses (middle), and featured figures and tables (right).

scMulti-omics analysis, the bipartite graph can be extended to a multipartite graph, where different modalities can be included as disjoint node types (e.g., genes, proteins, or peak regions). Such a multipartite heterogeneous graph can also include knowledge-based biological information, such as known molecular regulations and more than two modalities in one graph. However, by including more node types, the computational burden will be increased geometrically, which requires a dedicated discovery of model and parameter optimization in the future.

In summary, we evaluated DeepMAPS as a pioneer study for the integrative analysis of scMulti-omics data and cell-type-specific biological network inference. It will likely provide different visions of deep learning deployment in single-cell biology. With the development and maintenance of the DeepMAPS webserver, our long-term goal is to create a deep learning-based eco-community for archiving, analyzing, visualizing, and disseminating AI-ready scMulti-omics data.

# Methods

## Data description

We included ten public datasets (i.e., R-bench-1-3, C-bench-1-3, and A-bench-1-4) for grid-test benchmarking among DeepMAPS and existing tools and additional five datasets (i.e., R-test-1, C-test-1, and A-test-1-3) for independent test with optimized parameters. The human PBMC and lung tumor leukocyte CITE-seq data and 10× lymph node scRNA-seq & scATAC-seq data were used for the two case studies, respectively. All data are publicly available (Supplementary Data 1 and Data Availability).

## Data preprocessing and integration

The analysis of DeepMAPS takes the raw counts matrices of multiple scRNA-seq (multiple gene expression matrices), CITE-seq (gene and surface protein expressions matrices), and scRNA-ATAC-seq (gene expression and chromatin accessibility matrices) data as input. For each data matrix, we define modality representations as rows (genes, proteins, or peak regions) and cells as columns across the paper unless exceptions are mentioned. In each data matrix, a row or a column is removed if it contains less than 0.1% non-zero values. The data quality control is carried out by Seurat v3, including but not limited to total read counts, mitochondrial gene ratio, and blacklist ratio. Additional data preprocessing and integration methods are showcased below.

**Multiple scRNA-seq data.** Gene expression matrices are log-normalized, and the top 2000 highly variable genes are selected using Seurat v3[18] from each matrix. If there are less than 2000 genes in the matrix, all of them will be selected for integration. We then apply the widely-used canonical correlation analysis (CCA) in Seurat to align these matrices and harmonize scRNA-seq data, leading to a matrix $X = \{x_{ij} | i = 1,2,\ldots,I; j = 1,2,\ldots,J\}$ for $I$ genes in $J$ cells.

**CITE-seq data.** Gene and surface protein expression matrices are log-normalized. The top $I_1$ highly variable genes ($I_1 = 2000$) and $I_2$ proteins ($I_2$ is the total number of proteins in the matrix) are selected. The two matrices are then concatenated vertically, leading to a matrix $X = (x_{ij} | i = 1,2,\ldots,(I_1 + I_2); j = 1,2,\ldots,J)$ for $I_1$ genes and $I_2$ proteins in $J$ cells (can be treated as $I_1 + I_2 = I$ genes in $J$ cells). A centered log-ratio (CLR) transformation is performed on $X$ as follows:

$$CLR(x_{ij}) = \log\left(1 + \left(\frac{x_{ij}}{\exp\left(\frac{\sum_{i \in \mathscr{Z}_j} \log(1 + x_{ij})}{|\mathscr{Z}_j|}\right)}\right)\right) \quad (1)$$

where $\mathscr{Z}_j$ represents the set of indices for non-zero genes in cell $j$, and |•| means the number of elements in the set.

**scRNA-ATAC-seq data.** The gene expression matrix $X^R = \{x_{ij}^R | i = 1,2,\ldots,I; j = 1,2,\ldots,J\}$ with $I$ genes and $J$ cells is log-normalized. Then a left-truncated mixture Gaussian (LTMG) model is used to provide a qualitative representation of each gene over all cells, through the modeling of how underlying regulatory signals control gene expressions in a cell population[74]. Specifically, if gene $i$ can be represented by $G_i$ Gaussian distributions over all $J$ cells, that means there are potentially $G_i$ regulatory signals regulating this gene. A matrix $X^{R'} = \{x_{ij}^{R'}\}$ with the same dimension as $X^R$ can be generated, where the gene expressions are labeled by discrete values of $x_{ij}^{R'} = 1,2,\ldots G_i$.

The chromatin accessibility matrix is represented as $X^A = \{x_{kj}^A | k = 1,2,\ldots,K; j = 1,2,\ldots,J\}$ for $K$ peak regions in $J$ cells. We annotate peak regions in $X^A$ into corresponding genes based on the method described in MAESTRO[24]. Specifically, a regulatory potential weight $w_{ik}$ for peak $k$ to gene $i$ is calculated conditional to the distance of peak $k$ to gene $i$ in the genome:

$$w_{ik} = \begin{cases} 0, & d_{ik} > 150 \text{ kb or peak k located in any nearby genes} \\ \frac{1}{Length(exon)}, & \text{peak } k \text{ located at the exon regions of the gene } j \\ 2^{-\frac{d_{ik}}{d_0}}, & else \end{cases} \quad (2)$$

where $d_{ik}$ is the distance between the center of peak $k$ and the transcription start site of gene $i$, and $d_0$ is the half-decay of the distance (set to be 10 kb). The regulatory potential weight $w_{ik}$ of peak $k$ to gene $j$ is normally calculated by $2^{-\frac{d_{ik}}{d_0}}$. For peaks with $d_{ik} > 150kb$, $w_{ik}$ will be less than 0.0005, and thus we set it to 0 for convenience. In MAESTRO, for peaks located in the exon region, $d_0$ is 0, so that $w_{ik}$ should be 1 according to the formula, and $w_{ik}$ is normalized by the total exon length of gene $i$. The reason is that, in bulk ATAC-seq data, it is observed that many highly expressed genes will also have ATAC-seq peaks in the exon regions, mainly due to the temporal PolII and other transcriptional machinery bindings. Based on that observation, to better fit the model with gene expression, MAESTRO added the signals from the exon regions. However, as reads tend to be located in longer exons more easily than shorter exons, to normalize the possibility of background reads, it normalizes the total reads on exons by the total exon length for each gene. Eventually, a regulatory potential score of peak $k$ to gene $i$ in cell $j$ can be calculated as $r_{ik|j} = w_{ik} \times x_{kj}^A$. The scATAC-seq matrix $X^A$ can then be transformed into a gene regulatory potential matrix by summing up the regulatory potential scores of peaks that regulate the same gene:

$$x_{ij}^{A'} = 0 + \sum_k r_{ik|j}, \quad (3)$$

giving rise to the regulatory potential matrix $X^{A'} = \{x_{ij}^{A'} | i = 1,2,\ldots,I; j = 1,2,\ldots,J\}$ for same $I$ genes in $J$ cells with $X^R$.

We assume that the activity of a gene in a cell is determined by gene expression and gene regulatory activity with different contributions. Unlike the contribution weights determined directly based on the expression and chromatin accessibility values in Seurat v4 (weighted nearest neighbor)[5], we hypothesize that the relative contribution of the expression and chromatin accessibility of a gene to a cell is dynamic rather than static and not accurately determined with a snapshot of the cell. RNA velocity is determined by the abundance of unspliced and spliced mRNA in a cell. The amount of unspliced mRNA is determined by gene regulation and gene transcription rate, and the amount of spliced mRNA is determined by the difference between unspliced mRNA and degraded mRNA. We reasoned that for genes with positive RNA velocities in a cell, there are higher potentials to drive gene transcription. Thus, their regulatory activity related to chromatin accessibility has a greater influence than the gene expression in defining the overall transcriptional activity in the cell of the current snapshot. For genes with negative velocities, the transcription rate tends to be decelerated; hence chromatin accessibility has less influence on transcriptional activity than gene expression.

A velocity matrix $X^V = \{x_{ij}^V | i = 1,2,\ldots I; j = 1,2,\ldots,J\}$ is generated using scVelo with the default parameters[75]. Considering that some genes may fail to obtain valid velocity or regulatory potential values, we simultaneously remove the genes that have all-zero rows in $X^{A'}$ or $X^V$ from the four matrices $X^R, X^{R'}, X^{A'}, X^V$. Without loss of generality, we still use $I$ and $J$ represent the size of these new matrices. Furthermore, considering the potential bias when interpreting the velocity of a gene in a cell, we use the LTMG representations $x_{ij}^{R'} \in \{1,2,\ldots,G_i\}$ to discretize $x_{ij}^V$. For gene $i$, let $\mathcal{J}_g$ be the cell set where gene $i$ has the

same LTMG signal $g \in \{1, 2, \ldots, G_i\}$. For the cells in $\mathcal{J}_g$, we use the mean velocity of gene $i$ in these cells to replace the original velocities. To calculate a velocity weight $\beta$ of gene $i$ in cell $j$, we first extract $X_i^V = (x_{i1}^V, x_{i2}^V, \ldots x_{iJ}^V)$ for the velocity of gene $i$ in all cells and $X_j^V = (x_{1j}^V, x_{2j}^V, \ldots x_{Ij}^V)$ for the velocity of all genes in cell $j$. Then, for $X_i^V$, let $\mathcal{X}_i^{V+} = \{x_{ij}^V | x_{ij}^V > 0, j = 1, 2, \ldots J\}$ for all cells with positive velocities of gene $i$ and $\mathcal{X}_i^{V-} = \{x_{ij}^V | x_{ij}^V < 0, j = 1, 2, \ldots, J\}$ for all cells with negative velocities of gene $i$. Similarly, for $X_j^V$, let $\mathcal{X}_j^{V+} = \{x_{ij}^V | x_{ij}^V > 0; i = 1, 2, \ldots I\}$ for all genes with positive velocities in cell $i$ and $\mathcal{X}_j^{V-} = \{x_{ij}^V | x_{ij}^V < 0; i = 1, 2, \ldots I\}$ for all genes with negative velocities in cell $i$. For $x_{ij}^V > 0$, rank $\mathcal{X}_i^{V+}$ and $\mathcal{X}_j^{V+}$ from high to low based on velocity values with the ranking starting from 1 and calculate the velocity weight as:

$$\beta^+ = \sqrt{\left(|\mathcal{X}_i^{V+}| - a - 1\right)^2 + \left(|\mathcal{X}_j^{V+}| - b - 1\right)^2} \quad (4)$$

where $a$ is the rank of $x_{ij}^V$ in $\mathcal{X}_i^{V+}$, $b$ is the rank of $x_{ij}^V$ in $\mathcal{X}_j^{V+}$.

Similarly, for $x_{ij}^V < 0$, rank $\mathcal{X}_i^{V-}$ and $\mathcal{X}_j^{V-}$ from high to low based on absolute value of velocities with ranking starting from 0 and calculate the velocity weight as:

$$\beta^- = \sqrt{\mathcal{X}_j^{V-}| - a - 1)^2 + \left(|\mathcal{X}_j^{V-}| - b - 1\right)^2} \quad (5)$$

where $a$ is the rank of $x_{ij}^V$ in $\mathcal{X}_i^{V-}$ and $b$ is the rank of $x_{ij}^V$ in $\mathcal{X}_j^{V-}$.

We now generate a gene activity matrix $X^G = \{x_{ij}^G\}$, integrating gene expression and chromatin accessibility based on the velocity weight. $x_{ij}^G$ is the gene activity score (GAS) of gene $i$ in cell $j$:

$$x_{ij}^G = \begin{cases} x_{ij}^R + (1 + \beta^+) x_{ij}^{A'}, & \text{for } x_{ij}^V > 0 \\ x_{ij}^R + (1 - \beta^-) x_{ij}^{A'}, & \text{for } x_{ij}^V < 0 \\ x_{ij}^R + x_{ij}^{A'}, & \text{for } x_{ij}^V = 0 \end{cases} \quad (6)$$

### Construction of gene-cell heterogeneous graph

To simplify notations, we now redefine any integrative matrix generated in the previous section as $X = \{x_{ij} | i = 1, 2, \ldots, I; j = 1, 2, \ldots, J\}$ with $I$ genes and $J$ cells. $x_{ij}$ represents either normalized expressions (for multiple scRNA-seq and CITE-seq) or GAS (for scRNA-ATAC-seq) of gene $i$ in cell $j$. We calculate initial embeddings for genes and cells via two autoencoders. We used two autoencoders to generate the initial embeddings for cells and genes, respectively. The cell autoencoder reduces gene dimensions for each cell from I dimensions to 512 dimensions and eventually to 256 dimensions; a gene autoencoder reduces cell dimensions for each gene from J dimensions to 512 and 256 dimensions. So that, each cell and gene have the same initial embedding of 256 dimensions. The number of lower dimensions is optimized as a hyperparameter that differs for each dataset. The output layer is a reconstructed matrix $\hat{X}$ with the same dimension as $X$. The loss function of cell autoencoder is the mean squared error ($MSE$) of $X$ and $\hat{X}$:

$$loss = MSE(X, \hat{X}) = \sum_i (X - \hat{X})^2 \quad (7)$$

The gene autoencoder learns low dimensional features of genes from all cells, which has an encoder, latent space, and a decoder similar to the cell autoencoder, while the input $X^T$ is the transposed matrix of $X$. The loss function of gene autoencoder is

$$loss = MSE\left(X^T, \hat{X^T}\right) = \sum_j \left(X^T - \hat{X^T}\right)^2, \quad (8)$$

where $\hat{X^T}$ is the reconstructed matrix in the output layer with the same dimensions as $X^T$.

**Definition 1** (Heterogeneous graph): A heterogeneous graph is a graph with multiple types of nodes and/or multiple types of edges. We denote a heterogeneous graph as $G = (V, E, A, R)$, where $V$ represents nodes, $E$ represents edges, $A$ represents the node type union, and $R$ represents the edge type union.

**Definition 2** (Node type and edge type mapping function): We define $\tau(v) : V \rightarrow A$ and $\phi(e) : E \rightarrow R$ as the mapping function for node types and edge types, respectively.

**Definition 3** (Node meta relation): For a node pair of $v_1$ and $v_2$ linked by an edge $e_{1,2}$, the meta relation between $v_i$ and $v_j$ is denoted as $langle\tau(v_i), \phi(e_{i,j}), \tau(v_j)\rangle$.

Giving the integrated matrix $X$, we construct a bipartite gene-cell heterogeneous graph $G$ with two node types (cell and gene) and one edge type (gene-cell edge). $\mathbf{V} = \mathbf{V^C} \cup \mathbf{V^G}$, where $\mathbf{V^G} = \{v_i^G | i = 1, 2, \ldots, I\}$ denotes all genes, and $\mathbf{V^C} = \{v_j^C | j = 1, 2, \ldots J\}$ denotes all cells. $E = \{e_{i,j}\}$ represents the edge between $v_i^G$ and $v_j^C$. For $x_{ij} > 0$, the weight of the corresponding edge $\omega(e_{i,j}) = 1$, otherwise, $\omega(e_{i,j}) = 0$.

### Joint embedding via a heterogeneous graph transformer

We propose an unsupervised HGT framework[12,13] to learn graph embeddings of all the nodes and mine relationships between genes and cells. The input of HGT is the integrated matrix $X$, and the outputs are the embeddings of cells and genes and attention scores representing the importance of genes to cells.

**Definition 4** (Target node and source node): A node in $\mathbf{V}$ is considered as a target node, represented as $v_t$, when performing HGT to aggregate information and update embeddings of this node. A node is considered as a source node, represented as $v_s, v_s \neq v_t$, if there is an edge between $v_s$ and $v_t$ in $E$, denoted as $e_{s,t}$ for convenience.

**Definition 5** (Neighborhood graph of target node): A neighborhood graph of a target node $v_t$ is induced from $G$ and denoted as $G' = (V', E', A', R')$, where $\mathbf{V'} = \{v_t\} \cup \mathcal{N}(v_t)$, $\mathcal{N}(v_t)$ is the complete set of neighbors of $v_t$, $\mathbf{E'} = \{e_{i,j} \in E | v_i, v_j \in V'\}$, $A'$ marks the target and source node types, and $R'$ represents the target-source edge. $e_{s,t} \in E'$ represents the edge between $v_s$ and $v_t$. As only one edge type is included in $G$, the node meta relation of $v_s$ and $v_t$ is denoted as $\langle \tau(v_s), \phi(e_{s,t}), \tau(v_t) \rangle$.

1. Multi-head attention mechanism and linear mapping of vectors.
   Let $\mathcal{H}^\mathbf{l}$ denotes the embedding of the $l^{th}$ HGT layer ($l = 1, 2, \ldots, L$). The embedding of $v_t$ and $v_s$ on the $l^{th}$ layer is denoted as $\mathcal{H}^\mathbf{l}[\mathbf{v_t}]$ and $\mathcal{H}^\mathbf{l}[\mathbf{v_s}]$. A multi-head mechanism is applied to equally divide both $\mathcal{H}^\mathbf{l}[\mathbf{v_t}]$ and $\mathcal{H}^\mathbf{l}[\mathbf{v_s}]$ into $H$ heads. Multi-head attention allows the model to jointly attend to information from different embedding subspaces, and each head can run through an attention mechanism in parallel to reduce computational time. For the $h^{th}$ head in the $l^{th}$ HGT layer, the $\mathcal{H}^\mathbf{l}[\mathbf{v_t}]$ is updated from $\mathcal{H}^{\mathbf{l-1}}[\mathbf{v_t}]$ and $\mathcal{H}^{\mathbf{l-1}}[\mathbf{v_s}]$. The $\mathcal{H}^\mathbf{0}[\mathbf{v_t}]$ and $\mathcal{H}^\mathbf{0}[\mathbf{v_s}]$ are the initial embedding of $v_t$ and $v_s$, respectively. Three linear projection functions are applied to map node embeddings into the $h^{th}$ vector. Specifically, the $Q\_linear^h_{\tau(v_t)}$ function maps $v_t$ into the $h^{th}$ query vector $\mathbf{Q^h(v_t)}$, with dimension $\mathbb{R}^d \rightarrow \mathbb{R}^{\frac{d}{H}}$, where $d$ is the dimension of $\mathcal{H}^{\mathbf{l-1}}[\mathbf{v_t}]$ and $\frac{d}{H}$ is the vector dimension per head. Similarly, the $K\_linear^h_{\tau(v_s)}$ and $V\_linear^h_{\tau(v_s)}$ function map the source node $v_s$ into the $h^{th}$ key vector $\mathbf{K^h(v_s)}$ and the $h^{th}$ value vector $\mathbf{V^h(v_s)}$.

$$Q^h(v_t) = Q_{linear}{}^h_{\tau(v_t)}\left(\mathcal{H}^{(l-1)}[v_t]\right) \quad (9)$$

$$K^h(v_s) = K_{linear}{}^h_{\tau(v_s)}\left(\mathcal{H}^{(l-1)}[v_s]\right) \quad (10)$$

$$V^h(v_s) = V\text{-}linear^h_{\tau(v_s)}\left(\mathcal{H}^{(l-1)}[v_s]\right) \tag{11}$$

Each type of node has a unique linear projection to maximally model the distribution differences.

2. Heterogeneous mutual attention

To calculate the mutual attention between $v_t$ and $v_s$, we introduce the Attention operator which estimates the importance of each $v_s$ to $v_t$:

$$Attention(v_s, e_{s,t}, v_t) = \underset{\forall v \in \mathcal{N}(v_t)}{Softmax}\left(\underset{H}{\|}\left(ATT\_head^h(v_s, e_{s,t}, v_t)\right)\right) \tag{12}$$

The attention function can be described as mapping a query vector and a set of key-value pairs to an output for each node pair $e = (v_s, v_t)$. The overall attention of $v_t$ and $v_s$ is the concatenation of the attention weights in all heads, followed by a softmax function. $\underset{H}{\|}(\cdot)$ is the concatenation function. The ATT_head$^h(v_s, e_{s,t}, v_t)$ term is the $h^{th}$ head attention weight between the $v_t$ and $v_s$, which can be calculated by:

$$ATT\,head^h(v_s, e_{s,t}, v_t) = \left(K^h(v_s)W^{ATT}_{\phi(e_{s,t})}Q^h(v_t)^T\right) \\ \cdot \frac{\mu\langle\tau(v_s), \phi(e_{s,t}), \tau(v_t)\rangle}{\sqrt{d}}, \tag{13}$$

The similarity between the queries and keys was measured where $W^{ATT}_{\phi(e_{s,t})} \in \mathbb{R}^{\frac{d}{H} \times \frac{d}{H}}$ is a transformation matrix to capture meta-relation features. $(\cdot)^T$ is the transposal function and $\mu$ is a prior tensor to denote the significance for each the node meta relation $\langle\tau(v_s), \phi(e_{s,t}), \tau(v_t)\rangle$, serving as an adaptive scaling to the attention. The concatenation of attention heads results in the attention coefficients between $v_s$ and $v_t$, followed by a Softmax function in Eq. 12.

3. Heterogeneous message passing

A Message operator is used to extract the message of $v_s$ that can be passed to $v_t$. The multi-head *Message* is defined by:

$$Message(v_s, e_{s,t}, v_t) = \underset{H}{\|}\left(MSG\,head^h(v_s, e_{s,t}, v_t)\right) \tag{14}$$

The $h^{th}$ head message MSG_head$^h(v_s, e_{s,t}, v_t)$ for each edge $(v_s, v_t)$ is defined as:

$$MSG\,head^h(v_s, e_{s,t}, v_t) = V^h(v_s)W^{MSG}_{\phi(e_{s,t})} \tag{15}$$

where each source node $v_s$ in the head $h$ was mapped into a message vector by a linear projection $V^h(v_s) : \mathbb{R}^d \to \times\mathbb{R}^{\frac{d}{H}}$. $W^{MSG}_{\phi(e)} \in \mathbb{R}^{\frac{d}{H} \times \frac{d}{H}}$ is also a transformation matrix similar to $W^{ATT}_{\phi(e_{s,t})}$.

4. Target specific aggregation

To update the embedding of $v_t$, the final step in the $l^{th}$ HGT layer is to Aggregate the neighbor information obtained in this layer $\mathcal{H}^l[v_t]$ into the target node embedding $\mathcal{H}^{l-1}[v_t]$.

$$\widetilde{\mathcal{H}^l}[v_t] = \underset{\forall v_s \in \mathcal{N}(v_t)}{Aggregate}\left(Attention(v_s, e_{s,t}, v_t) \cdot Message(v_s, e_{s,t}, v_t)\right) \tag{16}$$

$$\mathcal{H}^l[v_t] = \theta\left(ReLU(\widetilde{\mathcal{H}^l}[v_t])\right) + (\theta - 1)\mathcal{H}^{l-1}[v_t], \tag{17}$$

where $\theta$ is a trainable parameter and *ReLU* is the activation function. The final embedding of $v_t$ is obtained by stacking information via all $L$ HGT layers, and $L$ is set to be 2 in DeepMAPS.

5. Determination of gene to cell attention

We call out the final attention score $\alpha_{ij}$ of gene $i$ to cell $j$ in the last HGT layer after the completion of the HGT process:

$$\alpha_{ij} = \sqrt{\sum_h ATT\_head^h(i,j)^2} \tag{18}$$

## HGT training on subgraphs

To improve the efficiency and capability of the HGT model on a giant heterogeneous graph (tens of thousands of nodes and millions of edges), we deploy a modified HGSampling method for subgraph selection and HGT training on multiple mini-batches[12]. For the graph $G$ with $I$ genes and $J$ cells, the union of subgraphs should cover $a\%$ (set to be 30%) nodes of gene and cell nodes to ensure the training power. As such, the sampler constructs a number of small subgraphs (50 in DeepMAPS) from the given heterogeneous graph $G$, and feeds the subgraphs into the HGT model in different batches using multiple GPUs. Each graph should include $a\% \times I/50$ genes, and $a\% \times J/50$ cells. Take a cell $j$ as a target node $v_t$ and its neighbor $v_s \in \mathcal{N}(v_t)$, corresponding to gene $i$, as source nodes, we calculate the probability on the edge $e_{s,t}$ as:

$$Prob(e_{s,t}) = \frac{x(v_s, v_t)}{\sum_{v \in \mathcal{N}(v_t)} x(v, v_t)}, \tag{19}$$

where $x(v_s, v_t) = x_{ij}$ refers to the expression or GAS value of the gene $i$ in cell $j$ in the integrated matrix $X$. Thus, for each target node $v_t$, we randomly select $\frac{a\% \times I/50}{a\% \times J/50}$ neighbor genes for $v_t$ based on sampling probability $\mathbf{Prob}(\mathbf{e_{s,t}})$. HGT hyperparameters, such as $W^{ATT}_{\phi(e_{s,t})}$, $W^{MSG}_{\phi(e_{s,t})}$, and $\theta$, will be trained and inherited sequentially from subgraphs 1 to 50 in one epoch. The subgraph training is performed in an unsupervised way with a graph autoencoder (GAE). The HGT is the encoder layer, and the inner product of embeddings is the decoder layer. We calculate the loss function of the GAE as the Kullback-Leibler divergence (**KL**) of reconstructed matrix $\hat{X}$ and the integrated matrix $X$:

$$loss = KL\left(softmax(\hat{X}), softmax(X)\right) \tag{20}$$

The subgraph training will be completed if the loss is restrained or reaches 100 epochs, whichever happens first.

## Determination of active genes module in cell clusters

**Predict cell clusters.** We deploy a Louvain clustering (Seurat v3) to predict cell clusters cell embeddings $\mathcal{H}^L[\mathbf{v_c}]$ generated from the final HGT layer. The resolution of Louvain clustering is determined by a grid-search test of multiple HGT hyperparameter combinations, and we set the clustering resolution of 0.4 as the default.

**Identify cell cluster-active gene association network.** We used an SFP model[17] to select genes that highly contribute to cell cluster characterization and construct cell cluster-active gene association networks. Define a new heterogeneous graph $\widetilde{G} = (V, \widetilde{E})$, $V \in V^G \cup V^C, \widetilde{E} \in \widetilde{E}^1 \cup \widetilde{E}^2$, where $\widetilde{E}^1$ represents the gene-gene relations, and $\widetilde{E}^2$ represents the gene-cell relations. The weight of the corresponding edge $\omega(\widetilde{e}^1_{i,i_2})$ of $v^G_{i_1} \in V^G$ and $v^G_{i_2} \in V^G$ is the Pearson's correlation of the HGT embeddings between $v^G_{j_1}$ and $v^G_{j_2}$. The weight of the corresponding edge $\omega(\widetilde{e}^2_{i,j})$ of $v^G_i \in V^G$ and $v^C_j \in V^C$ is the final attention score $\alpha_{ij}$. Only edges with $\omega(\widetilde{e}^1_{i_1,i_2}) > 0.5$ and

$\omega\left(\tilde{e}_{i,j}^2\right) > \mu\left(a_{i,j}\right) + sd(a_{i,j})$, where $\mu()$ represents the mean and $sd()$ represents the standard deviation of $a_{i,j}$, will be kept within a cell cluster. The weight of the remaining edges will then be max-min normalized to ensure an edge with the largest weight being rescaled to be 0 and an edge with the smallest weight being rescaled to be 1.

Let $Z$ be the number of clusters predicted via Louvain clustering, and $V^{C[z]} = \{v^{C[z]}\}$ be the node set corresponding with the cell set in cluster label of $z = 1,2,\ldots,Z$. We then formulate this problem using a combinatorial optimization model defined below

$$\min_{\substack{\tilde{E}^{\mathcal{L}} \subseteq \tilde{E}^1 \cup \tilde{E}^2}} \sum_{e \in \tilde{E}^{\mathcal{L}}} \omega(e)$$

s.t.

$$\mathcal{L}\left(v_{i_1}^C, v_{i_2}^C\right) = 1, \forall v_{i_1}^C, v_{i_2}^C \in V^{C[z]}, z = 1,2,\ldots,Z \qquad (21)$$

where $\mathcal{L}(v_{j_1}^C, v_{j_2}^C)$ is a binary indicator function representing whether two cell nodes, $v_{j_1}^C$ and $v_{j_2}^C$, can be connected (1) or not (0) in $\tilde{G}$ via a $\tilde{E}_{j_1,j_2}^{\mathcal{L}} = \{\tilde{e}_{i_1,j_1}^2, \tilde{e}_{i_1,i_2}^1, \tilde{e}_{i_2,i_3}^1 \ldots, \tilde{e}_{i_{t-1},i_t}^1, \tilde{e}_{i_t,j_2}^2\}$ path. Denote $\tilde{E}^{\mathcal{L}} = \{\tilde{E}_{j_1,j_2}^{\mathcal{L}}\}$ as the complete collection of $\tilde{E}_{j_1,j_2}^{\mathcal{L}}$ connecting $v_{j_1}^C$ and $v_{j_2}^C$. The combinatorial optimization model aims to identify the path connecting $v_{j_1}^C$ and $v_{j_2}^C$ with the minimum summed edge weight. We consider the gene networks remained in an SFP result of cluster $z$ as the cluster-active gene association networks.

## Construct GRNs from scRNA-ATAC-seq data

For genes in a cell cluster-active gene association network resulting from SFP, a set of TFs $q = 1,2,\ldots,Q$ can then be assigned to genes. The TF-peak relations are retrieved by finding alignments between the TF binding sites with peak regions in the scATAC-seq data, and the peak-gene relations are established previously when calculating the potential regulation scores $r_{ik|j}$ (Eq. 3). We design a regulatory intensity (RI) score $a_{i,j,q}$ to quantify the intensity of TF $q$ in regulating gene $i$ in the cell $j$:

$$a_{ij|q} = \sum_k b_{qk}^A \cdot r_{ik|j} \qquad (22)$$

where $b_{qk}^A$ is the binding affinity score of TF $q$ to peak $k$. The binding affinity score is calculated by three steps: (*a*) We retrieved the genome browser track file from JASPAR, which stores all known TF binding sites of each TF. A *p*-value score was calculated as $-\log_{10}(p) \times 100$ in JASPAR, where 0 corresponds to a *p*-value of 1 and 1,000 corresponds to a *p*-value $<10^{-10}$. We removed TF binding sites with *p*-value scores smaller than 500. (*b*) If a TF binding site overlaps with any peak regions in the scATAC-seq profile, it will be kept, otherwise, it will be removed. (*c*) Divide the corresponding *p*-value score by 100. We claim that a gene set regulated by the same TF is a regulon.

We calculate a regulon activity score (RAS) $\imath(q,z)$ of a regulon with genes regulated by TF $q$ in cell cluster $z$ as:

$$\imath(q,z) = \frac{\sum\limits_{i \in I_q} \sum\limits_{j \in C[z]} x_{ij} \cdot a_{ij|q}}{I \cdot J} \qquad (23)$$

where $I_q$ denotes genes regulated by TF $q$ in cell cluster $z$. We used the Wilcoxon rank-sum test to identify differentially active regulons in a cluster based on RAS. If the BH-adjusted *p*-value is less than 0.05 between different cell clusters and the log fold change larger than 0.10, we consider the regulon to be differentially active in this cluster, and it is defined as a cell-type-specific regulon (CTSR).

A GRN in a cell cluster is constructed by merging regulons in a cell cluster. The eigenvector centrality ($c_v$) of a TF node $v$ in GRN was defined as:

$$c_v = \alpha_{\max}(v) \qquad (24)$$

where $\alpha_{\max}$ is the eigenvector corresponding to the largest eigenvalue of the weighted adjacency matrix of a GRN. TFs with higher $c_v$ ranks were regarded as master TFs (top 10 by default).

## Benchmarking quantification and statistics

**Grid-search parameter test for cell clustering on benchmark data.** To determine the default parameters of HGT on different data types, we performed a grid-search test on HGT parameters, including the pair of number of embeddings and number of heads (91/13, 104/13, 112/16, and 128/16), learning rate (0.0001, 0.001, and 0.01), and training epochs (50, 75, and 100). Altogether, 36 parameter combinations were tested. For each of the three data types, the HGT parameter training were performed on three benchmark data, and the default parameter combination was selected based on the highest median score (ARI for multiple scRNA-seq data and CITE-seq data with benchmark labels and AWS for scRNA-ATAC-seq data without benchmark labels) of the three datasets.

To assess the performance of DeepMAPS alongside other proposed scMulti-omics benchmark tools, we compared DeepMAPS with Seurat (v3.2.3 and v4.0, https://github.com/satijalab/seurat), MOFA + (v1.0.0, https://github.com/bioFAM/MOFA2), Harmony (v0.1, https://github.com/immunogenomics/harmony), TotalVI (v0.10.0, https://github.com/YosefLab/scvi-tools), and GLUE (v0.3.2, https://github.com/gao-lab/GLUE). Because of the integration capability for different data types, DeepMAPS was compared with Seurat v 3.2.3 and Harmony on multiple scRNA-seq data, with Seurat v4.0.0, MOFA+, and TotalVI on CITE-seq data, and with Seurat v4.0.0, MOFA+, and GLUE on scRNA-ATAC-seq data. For each benchmarking tool, grid-search tests were also applied to a combination of parameters, such as the number of dimensions for cell clustering and clustering resolution.

The default HGT parameter combination selected for each data type was then applied to additional datasets (one multiple scRNA-seq, one CITE-seq, and three scRNA-ATAC-seq data) for independent tests. All benchmarking tools use their default parameters.

To showcase the rationale for selecting integrative methods and cell clustering methods in DeepMAPS, we evaluated the cell clustering performances by replacing the methods with several others. Specifically, for data integration, we replaced the CCA method with Harmony integration (multiple scRNA-seq), replaced the CLR method with Seurat weighted nearest neighbor method (CITE-seq), and replaced the velocity-weighted method with Seurat weighted nearest neighbor method and without using velocity (scRNA-ATAC-seq). For the cell clustering method, we replaced Louvain clustering with Leiden and the smart local moving (SLM) method. We also compared the influence of clustering resolution (use 0.4, 0.8, 1.2, and 1.6) to the cell clustering results in Deep-MAPS. Each comparison was performed on all 36 parameter combinations as used in the grid-search test. For DeepMAPS without velocity, we simply add up the gene expression matrix from scRNA-seq data and the gene potential activity matrix derived from scATAC-seq data, considering the balance weight introduced by velocity for gene $j$ in cell $i$ as 1.

**Adjusted rand index (ARI).** ARI is used to compute similarities by considering all pairs of the samples assigned in clusters in the current and previous clustering adjusted by random permutation. A contingency table is built to summarize the overlaps between the two cell label lists with $b$ elements (cells) to calculate the ARI. Each entry denotes the number of objects in common between the two label lists.

The ARI can be calculated as:

$$ARI = \frac{\frac{J_a + J_b}{C_n^2} - E\left[\frac{J_a + J_b}{C_n^2}\right]}{\max\left(\frac{J_a + J_b}{C_n^2}\right) - E\left[\frac{J_a + J_b}{C_n^2}\right]} \tag{25}$$

Where $E[.]$ is the expectation, $J_a$ is the number of cells assigned to the same cell cluster as benchmark labels; $J_b$ is the number of cells assigned to different cell clusters as benchmark labels; $C_n^2$ is the combination of selecting two cells from the total of $n$ cells in the cluster.

**Average Silhouette Width (ASW).** Unlike ARI, which requires known ground truth labels, a silhouette score refers to a method of interpretation and validation of consistency within clusters of data. The silhouette weight indicates how similar an object is to its cluster (cohesion) compared to other clusters (separation). The silhouette width ranges from −1 to +1, where a high value indicates that the object is well-matched to its cluster. The silhouette score sil(j) can be calculated by:

$$sil(j) = \frac{|n(j) - m(j)|}{\max\{m(j), n(j)\}} \tag{26}$$

where $m(j)$ is the average distance between a cell $j$ and all other cells in the same cluster, and $n(j)$ is the average distance of $i$ to all cells in the nearest cluster to which $j$ does not belong. We calculated the mean silhouette score of all cells as the ASW to represent the silhouette score of the dataset.

**Calinski-Harabasz index**. The CH index calculates the ratio of the sum of between-clusters dispersion and inter-cluster dispersion for all clusters. A higher CH index indicates a better performance. For a set of data E of size $n_E$ with $k$ clusters, the CH index is defined as:

$$CH = \frac{t(B_k)}{t(W_k)} \times \frac{n_E - k}{k - 1} \tag{27}$$

$$W_k = \sum_{q=1}^{k} \sum_{x \in C_q} \left(x - c_q\right)(x - c_q)^T \tag{28}$$

$$B_k = \sum_{q=1}^{k} n_q (c_q - c_E)(c_q - c_q)^T \tag{29}$$

where $t(B_k)$ is the trace of the between group dispersion matrix, and $t(W_k)$ is the trace of the within-cluster dispersion matrix. $C_q$ is the set of points in cluster $q$, $c_q$ is the center of cluster $q$, $c_E$ is the center of E, and $n_q$ is the number of points in cluster $q$. $T$ refers to the matrix transformation.

**Davies-Bouldin index**. The DB index signifies the average 'similarity' between clusters, where the similarity is a measure that compares the distance between clusters with the size of the clusters themselves. A lower DB index relates to a model with better separation between the clusters. For data with $k$ clusters, $i \in k$ and $j \in k$, the DB index is defined as:

$$DB = \frac{1}{k} \sum_{i=1}^{k} \max_{i \neq j} R_{ij} \tag{30}$$

$$R_{ij} = \frac{s_i + s_j}{d_{ij}} \tag{31}$$

where $s_i$ and $s_j$ are the average distance between each point within the cluster to the cluster centroid. $d_{ij}$ is the distance of cluster centroids of $i$ and $j$.

**Gene association network evaluations.** We evaluated the performance of the gene association network identified in DeepMAPS by comparing it with IRIS3[15] and a normal gene co-expression network inference using all genes. We calculated the closeness centrality and eigenvector centrality for the network generated in each tool. The formulations are given below.

**Closeness centrality (CC).** The closeness centrality (CC)[76] of a vertex $u$ is defined by the inverse of the sum length of the shortest paths to all the other vertices $v$ in the undirected weighted graph. The formulation is defined as:

$$CC(u) = \frac{1}{\sum_{v \in V} d_w(u, v)} \tag{32}$$

where $d_w(u, v)$ is the shortest weighted path between $u$ and $v$. If there is no path between vertex u and v, the total number of vertices is used in the formula instead of the path length. A higher CC indicates a node is more centralized in the network, reflecting a more important role of this gene in the network. The CC is calculated using igraph R package with function igraph::betweenness. We take the average CC of all nodes in a network to represent the network CC.

**Eigenvector centrality (EC).** Eigenvector centrality (EC)[77] scores correspond to the values of the first eigenvector of the graph adjacency matrix. The EC score of $u$ is defined as:

$$EC(u) = \lambda \sum_{v \in G} a_{uv} x_v \tag{33}$$

where $\lambda$ is inverse of the eigenvalue of eigenvector $x = (x_1, x_2, \ldots, x_n)$, $a_{uv}$ is the adjacent weighted matrix of undirect graph $G$. A node with a high eigenvector centrality score means that it is connected to many nodes which themselves have high scores. The EC is calculated using igraph R package with function igraph::evcent. We take the average EC of all nodes in a network to represent the network EC.

**Evaluations on GRN.** For scRNA-ATAC-seq data, we compared cell-type-specific GRNs inferred from DeepMAPS with (*i*) IRIS3 and SCENIC on the scRNA-seq matrix, (*ii*) IRIS3 and SCENIC on GAS matrix, (*iii*) MAESTRO on scATAC-seq matrix, and (*iv*) MAESTRO on original scRNA-seq and scATAC-seq matrix. For each dataset comparison, we set the cell clusters used in the benchmarking tool the same as generated in DeepMAPS to ensure fairness. GRNs generated from each tool were compared with three public functional databases, including Reactome[21], DoRothEA[22], and TRRUST v2[23]. Only human sample datasets were used for comparison as these databases are all human-related. We performed hypergeometric tests for GRN resulting in each tool to each database and compared the precision, recall, and F1 score of enriched GRNs and functional terminologies.

**Cell cluster leave-out test**
For a benchmark dataset with a real cell type label, we removed all cells in one cell type and ran DeepMAPS. Then, we traversed all cell types (one at a time) to evaluate the robustness of ARI. We removed cells in predicted cell clusters from DeepMAPS and other benchmark tools for data without benchmark labels.

**DeepMAPS server construction**
DeepMAPS is hosted on an HPE XL675d RHEL system with 2 × 128-core AMD EPYC 7H12 CPU, 64GB RAM, and 2×NVIDIA A100 40GB

GPU. The backend server is written in TypeScript using the NestJs framework. Auth0 is used as an independent module to provide user authentication and authorization services. Redis houses a queue of all pending analysis jobs. There are two types of jobs in DeepMAPS: The stateful jobs are handled by the Plumber R package to provide real-time interactive analysis; and the stateless jobs, such as CPU-bound bioinformatics pipelines and GPU training tasks that could take a very long time, are constructed using Nextflow. All running jobs are orchestrated using Nomad, allowing each job to be assigned with proper cores and storage and keeping jobs scalable based on the server load. The job results are deposited into a MySQL database. The front-end is built with NUXT, Vuetify as the UI library, Apache ECharts, and Cytoscape.js for data visualization. The frontend server and backend server are communicated using REST API.

## Reporting summary

Further information on research design is available in the Nature Portfolio Reporting Summary linked to this article.

## Data availability

All data used in this study are from public domain. The raw data are downloaded from the GEO database with the accession numbers for: human pancreatic islets scRNA-seq data GSE84133 and healthy bone marrow mononuclear cell CITE-seq data: GSE194122. The following datasets were obtained from figshare: the human pancreas scRNA-seq data [https://figshare.com/articles/dataset/Benchmarking_atlas-level_data_integration_in_single-cell_genomics_-_integration_task_datasets_Immune_and_pancreas_/12420968/8], the mouse bladder from the Tabula Muris scRNA-seq data [https://doi.org/10.6084/m9.figshare.5968960.v1], and the human lung adenocarcinoma PBMC CITE-seq data [https://doi.org/10.6084/m9.figshare.c.5018987.v1]. The following paired scRNA-seq and scATAC-seq datasets were obtained from the 10X Genomics website: 3k healthy PBMC data [https://www.10xgenomics.com/resources/datasets/pbmc-from-a-healthy-donor-no-cell-sorting-3-k-1-standard-2-0-0], 10k healthy PBMC data [https://www.10xgenomics.com/resources/datasets/pbmc-from-a-healthy-donor-granulocytes-removed-through-cell-sorting-10-k-1-standard-2-0-0], frozen human healthy brain data [https://www.10xgenomics.com/resources/datasets/frozen-human-healthy-brain-tissue-3-k-1-standard-2-0-0], 10k human PBMC data [https://www.10xgenomics.com/resources/datasets/10-k-human-pbm-cs-multiome-v-1-0-chromium-x-1-standard-2-0-0], healthy PBMC data [https://www.10xgenomics.com/resources/datasets/pbmc-from-a-healthy-donor-granulocytes-removed-through-cell-sorting-3-k-1-standard-2-0-0], fresh embryonic data [https://www.10xgenomics.com/resources/datasets/fresh-embryonic-e-18-mouse-brain-5-k-1-standard-2-0-0], and lymph node data [https://www.10xgenomics.com/resources/datasets/fresh-frozen-lymph-node-with-b-cell-lymphoma-14-k-sorted-nuclei-1-standard-2-0-0]. The scRNA-seq and scATAC-seq cancer cell line data was downloaded from CNGB Nucleotide Sequence Archive with an accession code of CNP0000213. All datasets are publicly available without restrictions. Details of data information can be found in Supplementary Data 1. Source data are provided with this paper.

## Code availability

The python source code of DeepMAPS Docker is freely available at https://github.com/OSU-BMBL/deepmaps and the DeepMAPS webserver is freely available at https://bmblx.bmi.osumc.edu/. The source code is also available on Zenodo[78].

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

## Acknowledgements

This work was supported by awards R01-GM131399 (Q.M.), R35-GM126985 (D.X.), and U54-AG075931 (Q.M.) from the National Institutes of Health. The work was also supported by the award NSF1945971 (Q.M.) from the National Science Foundation. This work was supported by the Pelotonia Institute of Immuno-Oncology (PIIO). The content is solely the responsibility of the authors and does not necessarily represent the official views of the PIIO. In addition, we thank Dr. Fei He from the Northeast Normal University for his valued suggestions in framework construction and data testing.

## Author contributions

Q.M., B.L., and D.X. conceived the basic idea and designed the framework. X.W. wrote the backbone code of DeepMAPS. C.W. and H.C. built the backend and front-end servers. S.G. designed the interactive figures on the server. Y.Liu carried out RNA velocity calculations. Y.Li designed the SFP model for gene module predictions. A.M, X.W., and J.L. carried out benchmark experiments. X.W., Y.C., and B.L. performed robustness tests. A.M., J.L., X.W., G.X., Z.L. and T.X. carried out the case study. J.W., D.W., Y.J., J.L., and L.S. performed tool optimizations. A.M. and Q.M. led the figure design and manuscript writing. All authors participated in the interpretation and writing of the manuscript.

## Competing interests

The authors declare no competing interests.
