## [Peer review file · Nature Communications]

REVIEWERS' COMMENTS

Reviewer #1 (Remarks to the Author):

After the latest round of revisions to address my comments, and the move to Nature Communications, I'm now content to support the publication of this manuscript.

Reviewer #3 (Remarks to the Author): Expert in computational cancer genomics, bioinformatics, machine learning, and single-cell genomics

The authors have addressed most of the comments rigorously, which significantly strengthened the manuscript. We really appreciate their efforts. However, we still have a few minor comments for their considerations to further improve the work:

1. As mentioned, in DeepMAPS, the authors used GNNs to generate gene & cell embeddings and multi-headed mechanism to prioritize key genes in cells, for which they have provided details in the methods. However, they should still provide a clearer intuitive explanation of this initial network creation and its impact – preferably in the overview.
2. As a related comment, Fig. 1 the manuscript is not clear/informative enough in this regard as opposed to the much clearer Supp Fig. 1.
3. The authors addressed the comment 2.1 (from the initial review) well, but have they incorporated a sentence summarizing the effect of data integration method to the main article? If not, please do it.
4. For comment 2.6a (from the initial review), we suggest the authors to add one sentence to clarify the motivation to apply CCA.
5. For Supp Fig. 7, we suggest the authors to group the performance plots of different methods according to the resolution value.
6. On lines 177-180, the authors emphasize that "Although for R-bench-1, A-bench-1, and A-bench-4, DeepMAPS did not achieve the best performance when comparing the median scores of all parameter combinations, it still held the advantage when comparing the highest score that a tool can achieve".

However, looking at Fig 2a, the results seem to show DeepMAPS only outperforms the other tools for the specific parameter settings mentioned. What is the reason for such discrepancies? Are these the median parameter settings for each tool or the best parameters?

7. On line 292, the term 'TF' is used for the first time without any formal definition. Please clarify what TF stands for here.

8. The methodological and results descriptions, though detailed enough, still are not sufficiently clear. For instance –

a. The embedding generation using autoencoders; however, the explanation provided in the reviewer response is much clearer.

b. On lines 349-353, the authors seem to be using CellChat to validate cell-cell interaction networks inferred from DeepMAPS. However, on line 434, they again mentioned using CellChat to generate the interaction networks from GAS matrix? Do you use CellChat to generate all these interaction networks or just validate your findings?

Reviewer #4 (Remarks to the Author): Expert in computational cancer genomics, bioinformatics, machine learning, and single-cell genomics. Co-reviewed with Reviewer #3 (same review).

Reviewer #5 (Remarks to the Author): Expert in computational cancer genomics, bioinformatics, machine learning, and single-cell genomics. Co-reviewed with Reviewer #3 (same review).

Reviewer #1 (Remarks to the Author):

After the latest round of revisions to address my comments, and the move to Nature Communications, I'm now content to support the publication of this manuscript.

Response: We appreciate the efforts of Reviewer #1 in evaluating our manuscript.

Reviewer #3 (Remarks to the Author): Expert in computational cancer genomics, bioinformatics, machine learning, and single-cell genomics

The authors have addressed most of the comments rigorously, which significantly strengthened the manuscript. We really appreciate their efforts. However, we still have a few minor comments for their considerations to further improve the work:

1. As mentioned, in DeepMAPS, the authors used GNNs to generate gene & cell embeddings and multi-headed mechanism to prioritize key genes in cells, for which they have provided details in the methods. However, they should still provide a clearer intuitive explanation of this initial network creation and its impact – preferably in the overview.

Response: We added one sentence in Lines 132-134 to explain the rationale of the initial heterogeneous network “Such a heterogeneous graph offers an opportunity to clearly represent and organically integrate scMulti-omics data so that biologically meaningful features can be learned synergistically”. We also explained, “The initial graph determines the paths of message passing and enables the calculation of attention scores in DeepMAPS”.

2. As a related comment, Fig. 1 the manuscript is not clear/informative enough in this regard as opposed to the much clearer Supp Fig. 1.

Response: We have replaced Fig. 1 with Supplementary Fig. 1.

3. The authors addressed the comment 2.1 (from the initial review) well, but have they incorporated a sentence summarizing the effect of data integration method to the main article? If not, please do it.

Response: Thanks for your suggestion. We have now added the sentence “*This integration process can ensure harmonizing datasets (especially for multiple scRNA-seq data) and generate an integrated matrix (with genes as rows and cells as columns) as the input for HGT*” in Lines 200-202.

4. For comment 2.6a (from the initial review), we suggest the authors to add one sentence to clarify the motivation to apply CCA.

Response: We have now added the clarification as “*We then apply the widely-used canonical correlation analysis (CCA) in Seurat to align these matrices and harmonize scRNA-seq data, leading to a matrix $X = \{x_{ij} | i = 1, 2, \dots, I; j = 1, 2, \dots, J\}$ for I genes in J cells.*” in Lines 526-528.

5. For Supp Fig. 7, we suggest the authors to group the performance plots of different methods according to the resolution value.

Response: We have regenerated Supplementary Fig. 7 as suggested.

6. On lines 177-180, the authors emphasize that "Although for R-bench-1, A-bench-1, and A-bench-4, DeepMAPS did not achieve the best performance when comparing the median scores of all parameter combinations, it still held the advantage when comparing the highest score that a tool can achieve". However, looking at Fig 2a, the results seem to show DeepMAPS only outperforms the other tools for the specific parameter settings mentioned. What is the reason for such discrepancies? Are these the median parameter settings for each tool or the best parameters?

Response: We apologize for the confusion. The sentence that the Reviewer quoted is based on our previous results in Revision round #1, which should be removed in the current version. Clearly, Fig. 2a showcased that "*DeepMAPS achieves the best performance comparing all benchmark tools in all test datasets in terms of ARI (for R-benches and C-benches) and ASW (for A-benches)*". We have now revised the conclusion in Lines 184-186. The parameter combination with the highest median ARI/ASW scores averaged in all benchmark datasets was considered as the default parameters for the corresponding data type.

7. On line 292, the term 'TF' is used for the first time without any formal definition. Please clarify what TF stands for here.

Response: Thanks for pointing this out. We have added the full name of TF in Line 283, where it first appears.

8. The methodological and results descriptions, though detailed enough, still are not sufficiently clear. For instance –

a. The embedding generation using autoencoders; however, the explanation provided in the reviewer response is much clearer.

Response: We have revised the explanation for autoencoders in Lines 623-628 based on our previous response:

"We used two autoencoders to generate the initial embeddings for cells and genes, respectively. The cell autoencoder reduces gene dimensions for each cell from I dimensions to 512 dimensions and eventually to 256 dimensions; a gene autoencoder reduces cell dimensions for each gene from J dimensions to 512 and 256 dimensions. So that, each cell and gene have the same initial embedding of 256 dimensions. The number of lower dimensions is optimized as a hyperparameter that differs for each dataset."

b. On lines 349-353, the authors seem to be using CellChat to validate cell-cell interaction networks inferred from DeepMAPS. However, on line 434, they again mentioned using CellChat to generate the interaction networks from GAS matrix? Do you use CellChat to generate all these interaction networks or just validate your findings?

Response: CellChat was used to generate all interaction networks from the integrated matrix (e.g., a GAS matrix); DeepMAPS does not have the function to infer cell-cell communications. We have now revised the description in Line 340-342 as “*Based on the cell types and raw data of gene and protein expressions, we inferred cell-cell communications and constructed communication networks among different cell types within multiple signaling pathways using CellChat (Fig. 4i)*”

Reviewer #4 (Remarks to the Author): Expert in computational cancer genomics, bioinformatics, machine learning, and single-cell genomics. Co-reviewed with Reviewer #3 (same review).

Response: We appreciate the efforts of Reviewer #4 in evaluating our manuscript.

Reviewer #5 (Remarks to the Author): Expert in computational cancer genomics, bioinformatics, machine learning, and single-cell genomics. Co-reviewed with Reviewer #3 (same review).

Response: We appreciate the efforts of Reviewer #5 in evaluating our manuscript.